

# A barycenter-based approach for the multi-model ensembling of subseasonal forecasts

Camille Le Coz[1], Alexis Tantet[1], Rémi Flamary[2], and Riwal Plougonven[1]

[1]LMD/IPSL, École Polytechnique, Institut Polytechnique de Paris, ENS, PSL Research University, Sorbonne Université, CNRS, Palaiseau France

[2]CMAP, Ecole Polytechnique, Institut Polytechnique de Paris, CNRS, Palaiseau, France

**Correspondence:** Camille Le Coz (camille.le-coz@lmd.ipsl.fr)

**Abstract.** Ensemble forecasts and their combination are examined from the perspective of probability spaces. Manipulating ensemble forecasts as discrete probability distributions, multi-model ensemble (MME) forecasts are reformulated as barycenters of these distributions. We consider two barycenters, each defined with respect to a different distance metric: the $L_2$ barycenter, which correspond to the traditional pooling method, and the Wasserstein barycenter, which better preserves certain geometric properties of the input ensemble distributions.

As a proof of concept, we apply the $L_2$ and Wasserstein barycenters to the combination of four models from the Subseasonal to Seasonal (S2S) prediction project database. Their performance is evaluated for the prediction of weekly 2m temperature, 10m wind speed, and 500hPa geopotential height over European winters. By construction, both barycenter-based MMEs have the same ensemble mean, but differ in their representation of the forecast uncertainty. Notably, the $L_2$ barycenter has a larger ensemble spread, making it more prone to under-confidence. While both methods perform similarly on average in terms of the Continuous Ranked Probability Score (CRPS), the Wasserstein barycenter performs better more frequently.

## 1 Introduction

**Multi-model ensemble methods (MME)**

Multi-model ensemble (MME) methods have been shown to improve forecast skill for different time scales, from short-range (Heizenreder et al., 2006; Casanova and Ahrens, 2009) and medium-range (Hamill, 2012; Hagedorn et al., 2012) to seasonal forecasting (Palmer et al., 2004; Alessandri et al., 2011; Kirtman et al., 2014). The added values of MME over single-model ensemble (SME) forecasts has been attributed to several factors. First, there is in general no "best" single-model (Hagedorn et al., 2005), the relative performances of the single-models vary depending on the considered target (i.e. region and variable of interest, metrics, etc.). The MME can take advantage of the complementary skill of the SMEs, thereby performing better on average. Second, Hagedorn et al. (2005) identified error cancellation and non-linearity of the metric as the main reasons for the





MME's performance exceeding the average performance of the SMEs. Third, MMEs allow us to explore a new dimension of uncertainty which remains unexplored by SMEs. Combining different models enables us to account for the uncertainty due to *model formulation*: by construction, a SME takes into account uncertainties in model initialization and can introduce variations in parameterizations to sample some of the uncertainty due to those parameterizations, but it cannot account for uncertainty due to model formulation. Besides, Weigel et al. (2008) showed that MME can improve the predictive skill if and only if the SMEs "fail to capture the full amount of forecast uncertainty". Finally, in addition, MMEs benefit from their larger ensemble size, which implies a better sampling of the real probability distribution.

**MME for subseasonal to seasonal (S2S) forecasts**

Subseasonal to seasonal forecasting bridges the gap between weather (medium range) and seasonal forecast. It corresponds to the time range between two weeks and up to two months. Predictions at this time scale are the focus of the Subseasonal to Seasonal (S2S) prediction project, whose objectives are to improve their skill and to promote their use (Vitart et al., 2017). As part of this project, a database containing S2S forecasts from twelve (originally eleven) operational centers has been made available to the research community. One of the main research questions the S2S project aims to answer is "What is the benefit of a multi-model forecast for subseasonal to seasonal prediction and how can it be constructed and implemented?" (Vitart et al., 2017).

Several studies have investigated the potential benefits of MME for S2S forecasts (Vigaud et al., 2017, 2020; Specq et al., 2020; Wang et al., 2020; Materia et al., 2020; Zheng et al., 2019; Pegion et al., 2019). These studies use different MME methods, variables and evaluation criteria, but they all concur that MMEs generally perform as well as, or better than, SMEs. Moreover, Vigaud et al. (2017) and Specq et al. (2020) suggest that MMEs not only improve the skill but also the reliability of the probabilistic forecasts. However, studies using the pooling method (i.e. the concatenation of the ensembles) point out that the better performance of the MME is also related to its larger number of members (Zheng et al., 2019; Specq et al., 2020; Wang et al., 2020). Among other studies, Karpechko et al. (2018) investigate a specific Sudden Stratospheric Warming event with a MME, while Ferrone et al. (2017) evaluates several MME methods, but neither compares the skill of the MME to that of the SMEs. Thus, while the potential benefits of MMEs for the subseasonal scale have been clearly established, the question of how to best combine ensemble forecasts from different models has received little attention. The present study focuses on this question.

The most direct and commonly used method for multi-model combination is the "pooling method", which simply concatenates the ensemble members from the different models. The members of the new multi-model ensemble can either have the same weights or be assigned different weights based on the model's skills (e.g. Weigel et al. (2008) for seasonal scale and Wanders and Wood (2016) for subseasonal scale). In the above mentioned studies, all used some variation of the pooling method except Vigaud et al. (2017, 2020) and Ferrone et al. (2017). The latter focused on the prediction of terciles, and so couple multi-model combination with other post-processing methods (e.g. model output statistics method such as extended logarithmic regression) to directly predict the terciles. Also focusing on quantiles, Gonzalez et al. (2021) used a sequential learning algorithm to linearly combine predictors (from the SMEs but also from the climatology and persistence), their weights being





updated at each step based on previous performances. Other methods for weighted-MME have been explored, but have not yet been applied to the subseasonal scale. At the seasonal scale, Rajagopalan et al. (2002) developed a Bayesian methodology to combine ensemble forecasts for categorical predictands (also used by Robertson et al. (2004) and Weigel et al. (2008) at the seasonal scale to obtain terciles of the MME). Other approaches, such as the Ensemble Model Output Statistics (EMOS) and

Bayesian Model Averaging (BMA), adopt a probability distribution perspective and aim to construct the PDF of the MME. In the EMOS method, an assumption is made on the shape of the PDF of the MME, and its parameters are then optimized with respect to a chosen score (e.g. the CRPS) on a training period (Gneiting et al., 2005). In contrast, in the BMA method, an assumption is made on the shape of the input distributions. The MME's PDF is then their weighted average, with the weights equal to the posterior probabilities of the input models (Raftery et al., 2005). The following framework of barycenter contains

the BMA as will be discussed later.

**MME as barycenters of discrete distributions**

In this study, we propose to revisit the combination of multiple model ensembles from a different perspective. We consider each ensemble of forecast as a discrete probability distribution and reformulate the multi-model ensemble as a barycenter of those distributions. We show that, in this framework, the pooled MME is actually the (weighted) barycenter with respect

to the $L_2$-distance. Since we work with distributions instead of a collection of ensemble members, the notion of barycenter can be extended to other metrics in the space of distributions. In particular, a natural distance in the distribution space is the Wasserstein distance that stems from the optimal transport theory (Villani, 2003). The Wasserstein distance is defined as the cost of the optimal transport between these two distributions.

  Optimal transport and the Wasserstein distance have been used in diverse applications for climate and weather. They have

been used to measure the response of climate attractors to different forcings (Robin et al., 2017), evaluate the performance of different climate models (Vissio et al., 2020), or assess different parametrizations (Vissio and Lucarini, 2018). Moreover, Papayiannis et al. (2018) used Wasserstein barycenters to point-downscale wind speed from an atmospheric model. Robin et al. (2019) developed a multivariate bias correction method based on optimal transport, while Ning et al. (2014) used it in the framework of data assimilation to deal with structural error in forecasts.

Here, we use the Wasserstein barycenters, or more precisely their Gaussian approximation, as a tool to build multi-model ensembles and compare it to the more traditional $L_2$-barycenter (i.e. the pooling method). The Wasserstein distance has some interesting properties in the probability distribution space. While the $L_2$ distance measures the "vertical" distance between the distributions, the Wasserstein distance is based on the "horizontal" displacement between them. This allowed the Wasserstein barycenter to retain some geometric properties of the input distributions (such as normality, for example, see Backhoff-

Veraguas, Julio et al. (2022)). We investigate the impact of changing the metric (from $L_2$ to Wasserstein) on the MME's performance by applying both barycenters to the combination of four models from the S2S database.

  The paper is organized as follows. The use of the two barycenters as MME methods is presented in Section 2. We first link the pooling method to the $L_2$-barycenter (2.2.1) before introducing the Wasserstein distance and its barycenter (2.2.2). We apply them to the combination of four models from the S2S database (namely ECMWF, NCEP, ECCC, and KMA; Vitart et al.,



2017). The case study is described in Section 3, including the datasets and evaluation metrics. The skill of the two MMEs and four SMEs are evaluated and compared in Section 4. The results are discussed in Section 5, and the main conclusions are highlighted in Section 6.

## 2  Multi-model ensemble methods

### 2.1  Ensemble forecasts as discrete probability distributions

At the S2S time scale, it is necessary to move from a deterministic to a probabilistic approach using ensemble forecasting (e.g. Kalnay (2003)). The ensemble is typically generated by perturbing the initial conditions and running the model for each of these perturbed states. The set of perturbed initial conditions represents the initial uncertainty associated with possible errors in the initial state of the atmosphere. This initial uncertainty is then transferred in time by the model. Thus, an ensemble forecast is a set of $N$ perturbed forecasts representing the evolution in time of the probability density of atmospheric variables according

to the model formulation.

An ensemble forecast aims at sampling the probability distribution of the forecasted variable conditioned on the initial distribution. Now, this can also be considered as a discrete probability distribution $\mu$ on the state space $\mathbb{R}^{n_t}$ such that

$$\mu = \sum_{i=1}^{N} a_i \delta_{\mathbf{x}_i}$$

where $N$ is the number of members in the ensemble, $\mathbf{x}_i \in \mathbb{R}^{n_t}$ is the position of the Dirac corresponding to the time-series of the $i$-th member, $n_t$ is the number of time steps, and $a_i$ is the weight of the $i$-th member (such that for all $i \leq n, a_i \geq 0$ and $\sum_{i=1}^{N} a_i = 1$). Thus, here, $\mathbf{x}_i$ does not represent an instantaneous state, but a full time series. That is, $\mu$ is a multivariate distribution where the $n_t$ variables correspond to the forecasted values at different lead-times. In a standard ensemble forecast,

all the members are equi-probable, so they have equal weights $a_i = \frac{1}{N}$. In the remainder of this section, we will look at ensemble forecast time-series from the angle of discrete distributions.

### 2.2  Barycenters for multi-model combination

The goal of multi-model ensemble methods can be rephrased as combining the complementary information from these distributions to obtain a new discrete distribution representing better the true probability distribution function of the forecasted

variable. A way to summarize a collection of distributions $(\mu_1, \mu_2, ..., \mu_K)$ is to compute their barycenter. The barycenter is found by solving the following minimization problem

$$\underset{\mu}{\arg\min} \quad \sum_{k=1}^{K} \lambda_k . d(\mu_k, \mu)^2 \tag{1}$$

where $\lambda_k$ represents the weights given to distributions, and $d(.,.)$ is a distance between distributions. The barycenter, also known as the Fréchet mean, is effectively the distribution that best represents the input distributions with respect to a criterion

given by the chosen distance $d$.





### 2.2.1 $L_2$ barycenter

The $L_2$ distance between two distributions $\mu_1$ and $\mu_2$ is given by $\|\mu_1 - \mu_2\|_2 = \left( \int_{\mathbb{R}^{n_t}} \left( \mu_1(\mathbf{z}) - \mu_2(\mathbf{z}) \right)^2 d^{n_t} \mathbf{z} \right)^{1/2}$. When using this distance in the barycenter equation (1), one can find the analytical formula for the $L_2$ barycenter of the distributions $\mu_1, ..., \mu_K$:

$$\mu_{L_2}^{\lambda} = \sum_{k=1}^{K} \lambda_k . \mu_k \tag{2}$$

$$= \sum_{k=1}^{K} \lambda_k . \sum_{i=1}^{N_k} a_{k,i} \delta_{\mathbf{x}_{k,i}} \tag{3}$$

where the distribution $\mu_k$ has $N_k$ Dirac located at points $(\mathbf{x}_{k,1}, ..., \mathbf{x}_{k,N_k}) \in \mathbb{R}^{n_t, N_k}$ with associated probability vector $\mathbf{a}_k = (a_{k,1}, ..., a_{k,N_k}) = (\frac{1}{N_k}, ..., \frac{1}{N_k})$ (for $k = 1, ..., K$).

Equation (2) shows that the $L_2$ barycenter is a weighted average of distributions. This is similar to the BMA method, which can then be seen as a $L_2$ barycenter with specific weights derived from the Bayesian framework (in which the weights are the posterior distributions of the input models, i.e. they are representing the probability that the associated model is the best model). However, Equation (3) differs from BMA in the particular distributions being summed : an assumption on their shape is made in the BMA, while the ensemble distributions are directly used in the $L_2$ barycenter (Raftery et al., 2005).

For a generic choice of weights $\lambda_k$, one can see that the $L_2$ barycenter corresponds to the concatenation of the members of the $d$ input ensembles with the re-scaling of the member's weights (using the weights $\lambda_k$, see Eq. (3)). This is equivalent to "pooling" together the ensemble's members from the different models. The pooling method is a simple and well-established MME method. It has been used for the seasonal scale (Hagedorn et al., 2005; Weigel et al., 2008; Robertson et al., 2004; Becker et al., 2014), for the decadal scale (Smith et al., 2013) and for the medium-range scale (Hamill, 2012; Hagedorn et al., 2012). More recently, it has also been applied on ensemble subseasonal forecasts by (Karpechko et al., 2018; Pegion et al., 2019; Zheng et al., 2019; Specq et al., 2020; Materia et al., 2020). It is interesting to note that the majority of MME methods for S2S forecasts use pooling and thus implicitly use the barycenter with the $L_2$ distance. However, the $L_2$ distance is not the only possible distance in the distribution space. In the following, we introduce another distance, the Wasserstein distance, and its associated barycenter.

### 2.2.2 Wasserstein barycenter

The Wasserstein distance stems from the optimal transport theory and can be seen as the cost of transportation between two distributions $\mu_1$ and $\mu_2$. It can be defined on discrete distributions as:

$$W_2^2(\mu_1, \mu_2) = \min_{\mathbf{T} \in U(\mathbf{a}_1, \mathbf{a}_2)} \langle \mathbf{T}, \mathbf{C} \rangle = \min_{\mathbf{T} \in U(\mathbf{a}_1, \mathbf{a}_2)} \sum_{i,j} t_{i,j} \|\mathbf{x}_{1,i} - \mathbf{x}_{2,j}\|^2 \tag{4}$$

where $U(\mathbf{a}_1, \mathbf{a}_2) = \left\{ \mathbf{T} \in \mathbb{R}_+^{N_1, N_2} : \mathbf{T}\mathbf{1}_{N_2} = \mathbf{a} \text{ and } \mathbf{T}^T \mathbf{1}_{N_1} = \mathbf{b} \right\}$ is the set of all the feasible transport matrices $\mathbf{T} = (t_{i,j})_{1 \le i \le N_1, 1 \le j \le N_2}$ between the probability vectors $\mathbf{a}_1$ and $\mathbf{a}_2$ associated with $\mu_1$ and $\mu_2$ respectively (with $\mathbf{1}_N$ standing for the all-ones vector



of size $N$), and $\mathbf{C} \in \mathbb{R}^{N_1 \times N_2}$ is a distance matrix whose elements are the pairwise squared euclidean distances between the elements of $\mathbf{x}_{1,:}$ and $\mathbf{x}_{2,:}$. Here, $(\langle \mathbf{T}, \mathbf{C} \rangle)^{1/2}$ is the cost associated with the transport $\mathbf{T}$ (with $\langle .,. \rangle$ being the Frobenius scalar product between matrices). The elements $t_{i,j}$ of a transport matrix $\mathbf{T}$ describe the amount of mass going from $\mathbf{x}_{1,i}$ to $\mathbf{x}_{2,j}$, while $\|\mathbf{x}_{1,i} - \mathbf{x}_{2,j}\|^2$ is the cost of moving one mass unit from $\mathbf{x}_{1,i}$ to $\mathbf{x}_{2,j}$. The minimization problem consists in searching for the optimal transport among all the feasible ones in $U(\mathbf{a}, \mathbf{b})$, i.e. the transport associated with the lowest cost denoted as the 2-Wasserstein distance. This is a short description of the Wasserstein distance for discrete distributions; for more information see Peyré and Cuturi (2020), Santambrogio (2015) or Villani (2003).

Contrary to the $L_2$ barycenter above, there is no general closed-form expression for the Wasserstein barycenter and one must solve the minimization problem (1) to find it. However, computing the Wasserstein barycenter directly on the discrete distributions causes an artificial decrease of the barycenter's variance, which leads to important under-dispersion of the multi-model ensembles. This variance shrinkage, illustrated in see appendix A, is due to the limited sample size of these distributions with respect to the space dimension.

In order to avoid the variance shrinkage and have a more efficient and better statistical estimation, we propose to estimate the barycenter using a Gaussian approximation of the data and then to "map" all the forecast onto this barycenter using the Gaussian mapping. That is, a Gaussian assumption is made to estimate a Gaussian Wasserstein barycenter which provides in closed form an affine mapping from each distribution to the barycenter, and then this mapping is applied to the discrete distributions which aligns them on the barycenter but keep their individual samples. This means that only the first and second order moments of the Gaussian are estimated from the data which attenuates the statistical the difficulty of estimating the Wasserstein distance Flamary et al. (2020). This approach was first used in Gnassounou et al. (2023) for data normalization of biomedical signals. Note that they additionally assumed the signal to be stationary and periodic which we did not do. The method is illustrated in Figure 1 and is made up of four steps:

1. Each discrete distribution $\mu_k$ is approximated by a multivariate normal distribution $\hat{\mu}_k$ of mean $\hat{\mathbf{m}}_k$ and covariance $\hat{\boldsymbol{\Sigma}}_k$ (for $k = 1, \ldots, K$), which are estimated by the unbiased sample mean and covariance respectively.

2. The Wasserstein barycenter between the normal distributions is computed. It is also a multivariate normal distribution with mean $\hat{\mathbf{m}}_b = \sum_{k=1}^{K} \lambda_k \hat{\mathbf{m}}_k$ and covariance $\hat{\boldsymbol{\Sigma}}_b$ that is estimated iteratively using the fixed-point iteration $\hat{\boldsymbol{\Sigma}}_b = \sum_{k=1}^{K} \lambda_k \left( \hat{\boldsymbol{\Sigma}}_b^{1/2} \hat{\boldsymbol{\Sigma}}_k^{1/2} \hat{\boldsymbol{\Sigma}}_b^{1/2} \right)^{1/2}$ from Agueh and Carlier (2011).

3. The optimal transports from each input Gaussian distribution $\hat{\mu}_k$ to the Gaussian barycenter $\hat{\mu}_b$ are derived. They are given by the following affine mappings $T_k(\mathbf{x}) = \mathbf{A}_k(\mathbf{x} - \hat{\mathbf{m}}_k) + \hat{\mathbf{m}}_b$ with $\mathbf{A}_k = \hat{\boldsymbol{\Sigma}}_k^{-1/2} \left( \hat{\boldsymbol{\Sigma}}_k^{1/2} \hat{\boldsymbol{\Sigma}}_b \hat{\boldsymbol{\Sigma}}_k^{1/2} \right)^{1/2} \hat{\boldsymbol{\Sigma}}_k^{-1/2}$ (assuming that the $\Sigma_k$ are invertible, see below).

4. The mappings are applied to the corresponding discrete distributions (that is the mapping $T_k$ is applied to each state members $\mathbf{x}_{k,i}$ of $\mu_k$), and then the adapted distributions are pooled together (as for the $L_2$ barycenter in eq. (3)) to create the 2-Wasserstein barycenter with Gaussian mapping ($GaussW_2$ barycenter hereafter).





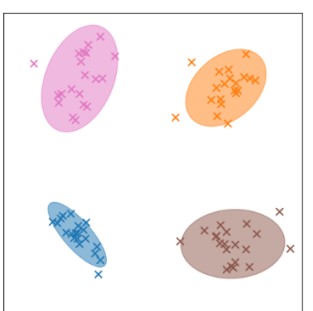 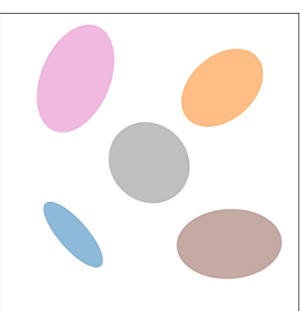 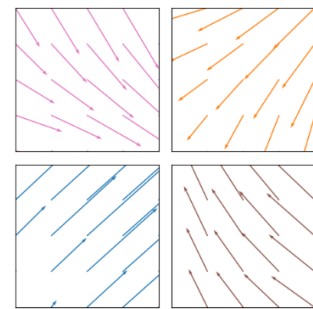 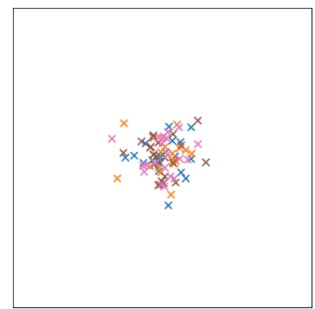

**Step 1:** Approximation of the discrete distributions (crosses representing member states) by Gaussians (shadings representing an isoline of density).

**Step 2:** Computation of the Wasserstein barycenter of the Gaussian distributions (in grey).

**Step 3:** Computation of the continuous mappings of each Gaussian distribution to the barycenter (arrows representing the displacement).

**Step 4:** Application of the mappings on the discrete distributions and pooling (the colors indicating the original distribution of the state members before pooling).

**Figure 1.** Illustration of the Wasserstein barycenter with Gaussian mapping approach.

This approach computes mappings and barycenters on continuous (multivariate normal) distributions, thereby avoiding variance shrinkage of the barycenters due to limited discrete sample sizes (see Appendix A). Computing the mean and covariance of a discrete distribution for the multivariate normal approximation is less sensitive to the sampling of the distributions (Gnassounou et al., 2023), and regularization can be used if needed to ensure the invertibility of the covariance matrices $\hat{\Sigma}_k$. This is especially important when the number of members (i.e. samples) is small compared to the number of dimensions. The approximation of the discrete distributions by multivariate normal distributions is a strong hypothesis hypothesis considering that ensembles in climate science are transferred according to nonlinear evolution equations. However, this is a classical hypothesis for modeling stochastic processes and in signal processing. Moreover, this hypothesis is only used to derive Gaussian barycenter and the mappings which are then applied to the original discrete distributions which can be seen as a second order calibration of the models onto a barycenter which preserves higher order moments through the individual samples.

Another advantage of this approach is the computational cost (in low dimensions). Estimating the parameters of the multivariate normal distribution from the discrete one has a computational complexity of $\mathcal{O}(Nd^2)$ (where $N$ is the number of points in the discrete distribution, and $d$ the dimension of the sample space, here $d = n_t$). The Gaussian mappings and Gaussian barycenter computation have a complexity of $\mathcal{O}(d^3)$ and $\mathcal{O}(d^3 log(1/\epsilon))$ respectively (where $\epsilon$ is the tolerance error in the fixed-point algorithm). On the other hand, computing the exact barycenter requires multiple resolution (per model and iteration) of exact OT problems $\mathcal{O}(N^3 log(N))$ (Agueh and Carlier, 2011).




### 2.2.3 Illustration of the application of barycenters to ensemble forecasts

The $L_2$ and $W_2$ distances have different properties in the distribution space, leading to distinct multi-model ensembles. The
$L_2$ distance focuses on the "vertical" differences between the distributions. This means that the $L_2$ distance between two
distributions with disjoint supports is equal to the sum of their $L_2$ norms, no matter how distant their supports are (e.g. it is
equal to $2\sum_i^N 1/N^2$ for two discrete distributions with both $N$ equi-probable points). As a result, the members of the L2-
barycenter always coincide with the members of the initial ensembles, regardless of the distance between the supports of the
latter. In contrast, the $GaussW_2$ distance is a "horizontal" measure of the difference between two distributions, in the sense
that it is based on the horizontal displacement (or mapping) between them (Santambrogio, 2015). This allowed the associated
barycenter to have a different support than the input distributions. That is, the members of the $GaussW_2$ barycenter do not
coincide with the members of the initial ensembles.

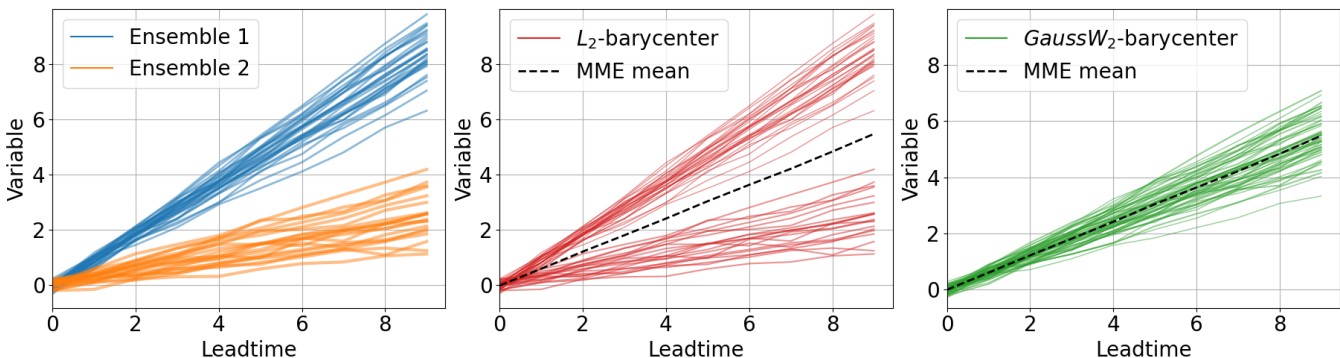

**Figure 2.** Illustration of (equal-weights) multi-model ensembles using barycenters for a synthetic variable. The ensemble can be seen as a
discrete distribution whose points are the time series of each member. The weight of the points in the distribution is indicated here by the
thickness of the line.

Figure 2 gives an illustration of the two barycenters applied to the combination of two synthetic ensemble forecasts with
$N_1 = 10$ and $N_2 = 15$ members respectively . The $L_2$ barycenter in Figure 2b is the concatenation of the two input ensembles
in Figure 2a. The weight of each member of the $L_2$ barycenter is given by the weight of the corresponding member in the input
ensemble ($1/N_1$ or $1/N_2$) multiplied by the corresponding model's weight (here $\lambda_1 = \lambda_2 = 0.5$). The $GaussW_2$ barycenter
shown in Fig. 2c has a different structure compared to the $L_2$ barycenter. As expected, a major difference is that none of its
members belongs to the input ensembles. In other words, while the support of the $L_2$ barycenter is the union of the input sup-
ports, the support of the $GaussW_2$ barycenter lies along the path of the optimal transport and can therefore be different. This
also allows for the $GaussW_2$ barycenter to retain certain characteristics of the input distributions. For example, in this illustra-
tive case, both input distributions are unimodal, and so is the $GaussW_2$ barycenter. This is not the case for the $L_2$ barycenter
which has two modes, each associated with the mode of one of the input distributions. The two barycenters may be attractive
for different types of forecasts. The $GaussW_2$ barycenter preserves some geometric properties of the input distributions, while
the $L_2$ barycenter retains bimodality, which may be of interest (Bertossa et al., 2023). Actual forecasts are more complex, and



an assessment of the different barycenters will require extensive testing with different criteria. The purpose of Figure 2 is to emphasize the (hitherto largely untapped) variety of ways to build multi-model ensemble forecasts.

An important fact to note is that, despite their differences, the two barycenters have the same ensemble means (for the same model weights). Their mean is equal to the weighted mean of the input distribution's means (this follow easily from Eq. (3) for the $L_2$ barycenter, and see Peyré and Cuturi (2020, Remark 9.1) for the $W_2$ and $GaussW_2$ barycenters). Thus, the difference

between the barycenters is the way they represent the forecast uncertainty. This raises the question: how different are the $L_2$ and $GaussW_2$ distributions and which one captures better the forecast uncertainty? We address this question empirically for a particular case study in the next sections.

## 3    Data and methodology

In this study, we focus on subseasonal forecasts over Europe during boreal winter. We consider one large-scale and two surface

variables: the geopotential height at 500hPa (z500), the daily temperature at 2m (t2m), and the wind speed at 10m (w10m). The latter two variables are interesting to predict for the energy sector, as the energy demand is particularly dependent on the temperature in winter (due to the use of electrical heating), while the wind speed is an essential variable for estimating wind farm production. The geopotential height at 500 hPa is included as a traditional large-scale variable for comparison.

### 3.1    Data

To make predictions for this case study and to validate them, we need both a dataset of ensemble forecasts from multiple dynamical models, to which the barycenters will be applied, and a reference dataset against which the forecast skills will be evaluated.

### 3.1.1    S2S data

For this first implementation, we selected four models from the S2S database (Vitart et al., 2017): the European Centre for

Medium-Range Weather Forecasts (ECMWF) model, the National Centers for Environmental Prediction (NCEP) model, the Environment and Climate Change Canada (ECCC) model, and the Korea Meteorological Administration (KMA) model. An important criterion in model selection is their diversity. As previously explained, one advantage of doing multi-model ensemble is to better sample the model error. If the models have similar construction, this error is not sampled properly. Others criteria include time coverage and consistency in production over the study period.

The forecasts and reforecasts from the selected models were retrieved from the S2S database through the ECMWF's Meteorological Archival and Retrieval System (MARS) directly on a $1.5° \times 1.5°$ grid for our study domain shown in Figure 3, that is Europe ($34°N - 74°N$, $13°W - 40°E$). We retrieved the daily mean 2m temperature, the instantaneous values of the 10m wind speed, and the 500hPa geopotential height at midnight (00Z). We selected the forecasts initialized during the months of December-January-February (DJF) for the 2016–2022 period, for which the four models have matching starting dates. This

resulted in a total of 90 simulations spanning seven winters.





**Table 1.** Description of the four models from the S2S database (characteristics for the period used). See Vitart et al. (2017) for more details.

|  | ECMWF | NCEP | ECCC | KMA |
|---|---|---|---|---|
| **Forecasts** | | | | |
| Time range | d 0–46 | d 0–44 | d 0–32 | d 0–60 |
| Resolution | Tco639/Tco319 L137 | T126 L64 | Yin-Yang grid at 35° (varying vertical resolution) | N216 L85 |
| Frequency | Twice a week (Monday, Thursday) | Daily | Weekly (Thursday) | Daily |
| Ensemble size (original) | 50+1 | 15+1 | 20+1 | 4 |
| Ensemble size after pre-processing | 51 | 48 | 21 | 32 |
| **Reforecasts** | | | | |
| Method | On the fly | Fixed | On the fly | On the fly |
| Period | Past 20 years | 1999–2010 | fixed period (varying) | fixed period (varying) |
| Frequency | Twice a week | Daily | Weekly | 4/month |
| Ensemble size | 10+1 | 3+1 | 3+1 | 3 |

**Pre-processing:**

– Due to model errors, forecasts tend to drift away from the observed climate toward the model climatology as lead time increases (Takaya, 2019). Therefore, it is important to calibrate extended-range forecasts. The model climatology is estimated from reforecasts, which are "retrospective" forecasts run generated using the same model as the forecast but for past periods. The observed climatology is derived from a reference dataset (see Section 3.1.2 below). Then, calibration methods are applied to statistically correct the forecasts. Here, we use the Mean and Variance Adjustment (MVA, Leung et al. (1999); Manzanas et al. (2019)) method, following Goutham et al. (2022). To ensure a robust estimation of the climatology, we aggregate all forecasts initialized within a 15 days window, including the same calendar day as the forecast starting date and the previous 14 days[1] (Manrique-Suñén et al., 2020). The observed climatology is computed on the same dates for consistency.

– The NCEP and KMA models have smaller ensemble sizes but are produced daily. We thus create a lagged ensemble by aggregating the forecasts with the ones initialized the two preceding days for NCEP and the preceding seven days for KMA. This allows us to improve their skills and to increase their ensemble size, making them more comparable to

---

[1]previous 15 days for KMA to better fit the frequency of its reforecasts



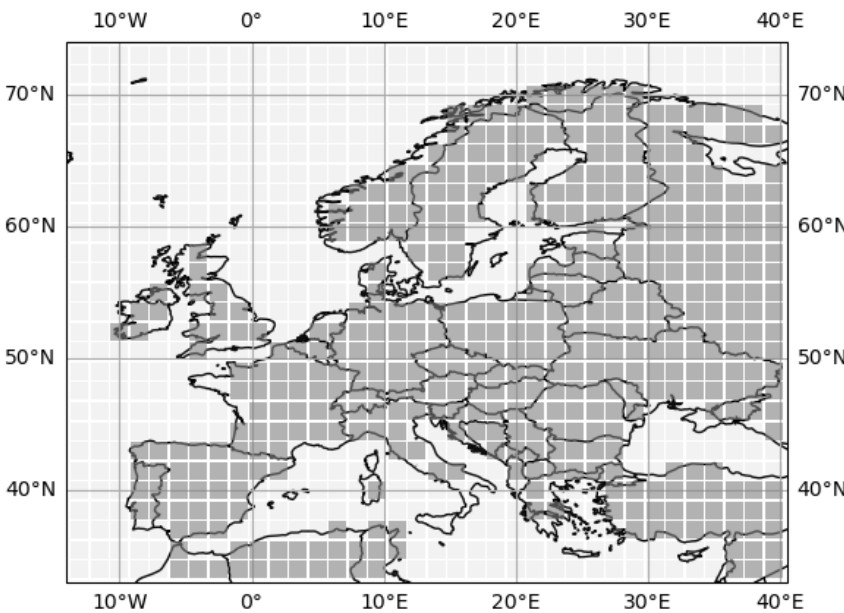

**Figure 3.** Study domain with the $1.5° \times 1.5°$ lat/lon grid from the forecasts. Only the land points (in grey) are used in this case study.

the other models (48 and 32 members for NCEP and KMA, respectively after lagging). The ECCC model has a smaller ensemble size than ECMWF and the lagged NCEP and KMA models. However, lagging is not applied to it because of its frequency (it is only produced weekly).

### 3.1.2 Reference data

The Modern-Era Retrospective analysis for Research and Applications, version 2 (MERRA-2, Gelaro et al. (2017)) reanalysis is used as reference for calibration and forecast validation. We use a reanalysis as reference in order to have a spatially and temporally complete dataset. MERRA-2 reanalysis was chosen because it is a recent reanalysis covering both the calibration and validation periods and is based on a different circulation model from those used in the selected S2S models. This would not be the case, for example, with the ERA5 reanalysis which is also produced by the ECMWF with a similar model as the S2S forecasts (Hersbach et al., 2020). The three variables of interest were retrieved from NASA Goddard Earth Sciences (GES) Data and Information Services Center (DISC) on MERRA-2's native grid ($0.5°$ lat $\times$ $0.625°$ lon grid). The data was re-gridded to match the $1.5° \times$ $1.5°$ lat/lon grid of the forecasts using bi-linear interpolation with the Climate Data Operators (CDO, Schulzweida (2022)).

We also use MERRA-2 to build a 30-year rolling climatology. The climatology is a common benchmark for forecast validation, and it is also used to compute skill scores (see Section 3.2). For a given day of the year, the climatology consist of



MERRA-2 values for the same day over the previous 30 years. The climatology is thus an ensemble with 30 members, each

corresponding to a different year.

## 3.2  Evaluation

We now discuss the different evaluation scores that are used in this paper such as scores based on CRPS and the Spread-Skill Ratio more related to calibration. The scores are detailed in Table 2.

### 3.2.1  Continuous Ranked Probability Score (CRPS)

The CRPS is a widely used score for probabilistic forecasts of continuous variable (Matheson and Winkler, 1976; Gneiting and Raftery, 2007; Wilks, 2019). The CRPS is actually the $L_2$ distance between the Cumulative Density Function (CDF) of the ensemble forecast and the CDF of the observation:

$$\text{CRPS}_n = \int\limits_{-\infty}^{+\infty} \left[ F_{\text{fc},n}(y) - F_{\text{ref},n}(y) \right]^2 dy \tag{5}$$

where $F_{\text{fc}}$ is the CDF of the forecast , $F_{\text{ref}}$ is the CDF of the observation and $n \in [\![1, N]\!]$ is the simulation's number. The CDFs

are computed empirically from the ensembles. In general the reference is deterministic, and its CDF is then a step function $F_{\text{ref}}(y) = \mathbf{1}_{y \geq x_{\text{ref}}}$, where $x_{\text{ref}}$ is the deterministic value of the reference.

The accuracy of the ensembles is evaluated by computing the mean CRPS (over the forecasts $n$). For easier comparison across variables, we use the Continuous Ranked Probability Skill Score (CRPSS) that compare the mean CRPS of the ensemble forecast to the mean CRPS of the climatology, but the statistical tests will be performed on the CRPS distribution.

We consider two other scores based on the CRPS. To evaluate robustness to outliers, we use the proportion of skillful forecasts (hereafter CRPSp), which corresponds to the percentage of simulations for which the model has a better CRPS than the climatology (Goutham et al., 2022). We consider that the model has some skill if 50% of the forecasts are skillful. As a counter-part, we introduce a third score that focuses on the tail of the distribution (i.e. forecasts far from the truth). The proportion of critical failures (CRPSf) is defined as the percentage of simulations for which the model has a $\text{CRPS}_n$ larger than

twice that of the climatology (this metric is thus negatively oriented contrary to the CRPSS and CRPSp).

**Remarks** on the links between the CRPS and the $L_2$ barycenter:

  – It is interesting to note that for univariate distribution, the $L_2$ barycenter is also the barycenter with respect to the energy distance (see appendix B). Thus, since the CRPS is identical to the energy distance in 1D (see Wilks, 2019), we can expect a good CRPS performance for the $L_2$ barycenter.

For symmetry, the Wasserstein distance could also be used as a score for forecast validation. However, it is equivalent to the RMSE (averaged over the members) in the case of a deterministic observation and so it does not evaluate well the representation of the uncertainties in the forecasts. Thus, we do not use it here.





– The CRPS of the $L_2$ barycenter $\mu_{L_2}^\alpha$ of the distributions $(\mu_1,\ldots,\mu_d)$ (with weights $\lambda_1,\ldots,\lambda_d$) can be expressed as a function of the their CRPS to the observation and cross-CRPS:

$$\text{CRPS}(\mu_{L2},\nu_{obs}) = \sum_{i=1}^{n}\lambda_i\text{CRPS}(\mu_i,\nu_{obs}) - \sum_{i=1}^{n}\sum_{j\neq i}\frac{1}{2}\lambda_i\lambda_j\text{CRPS}(\mu_i,\mu_j) \tag{6}$$

where $\nu_{obs}$ is the observation (see appendix C). This expression clearly shows that the CRPS of the $L_2$ barycenter is always smaller than the average CRPS of the input distributions. In fact, the more different are the input distributions (i.e. large cross-CRPS), the more advantageous it is to merge them.

### 3.2.2 Spread-skill ratio

The CRPS evaluates the accuracy of the forecasts, that is, their overall quality (Wilks, 2019). In order to focus on the calibration (or reliability) of the ensembles, we also look at the spread-skill ratio (SSR) defined as the ratio of the ensemble standard deviation over the root mean square error RMSE of the ensemble mean. A perfectly reliable ensemble has a SSR equal to one, while larger SSR indicates overdispersion and smaller values underdispersion (Fortin et al., 2014). The SSR can be used as a first evaluation of an ensemble's reliability (a value of 1 is necessary, though not sufficient, for a fully reliable ensemble.).

**Table 2.** Scores used for evaluation.

| Name | Formula | Range | Comment |
|---|---|---|---|
| $\overline{\text{CRPSm}_l}$ | $\dfrac{1}{N_{for}}\displaystyle\sum_{k=1}^{N_{for}}\dfrac{1}{N_{lat}N_{lon}}\sum_{i=1}^{N_{lat}}\sum_{j=1}^{N_{lon}}w_i\text{CRPS}_{\text{fc},k,i,j,l}$ | $[0,+\infty)$ | negatively oriented |
| CRPSS | $1-\dfrac{\overline{\text{CRPSm}_l}(\text{fc})}{\overline{\text{CRPSm}_l}(\text{clim})}$ | $(-\infty,1]$ | positively oriented |
| CRPSp | $\dfrac{1}{N_{for}}\displaystyle\sum_{k=1}^{N_{for}}\dfrac{100}{N_{lat}N_{lon}}\sum_{i=1}^{N_{lat}}\sum_{j=1}^{N_{lon}}[\text{CRPS}_{\text{fc},k,i,j,l}<\text{CRPS}_{\text{clim},k,i,j,l}]$ | $[0,100]$ | positively oriented |
| CRPSf | $\dfrac{1}{N_{for}}\displaystyle\sum_{k=1}^{N_{for}}\dfrac{1}{N_{lat}N_{lon}}\sum_{i=1}^{N_{lat}}\sum_{j=1}^{N_{lon}}[\text{CRPS}_{\text{fc},k,i,j,l}>2\,\text{CRPS}_{\text{clim},k,i,j,l}]$ | $[0,100]$ | negatively oriented |
| SSR | $SSR=\dfrac{\sqrt{\frac{1}{NN_{lat}N_{lon}}\sum_{n=1}^{N}\sum_{i=1}^{N_{lat}}\sum_{j=1}^{N_{lon}}w_i\text{var}_m(\text{fc}_{n,i,j,m})}}{\sqrt{\frac{1}{NN_{lat}N_{lon}}\sum_{n=1}^{N}\sum_{i=1}^{N_{lat}}\sum_{j=1}^{N_{lon}}w_i(\text{mean}_m(\text{fc}_{k,i,j,l,m})-\text{obs}_{k,i,j})^2}}$ | $[0,+\infty)$ | SSR$>1$: overdispersion SSR$<1$: underdipersion |
| Notations | $i\in[\![1,N_{lat}]\!]$: latitude; $j\in[\![1,N_{lon}]\!]$: longitude; $l\in[\![1,N_{day}]\!]$: lead time; $k\in[\![1,N_{for}]\!]$: forecast (i.e. starting date); $m\in[\![1,N_{mem}]\!]$: member. | | |
| Remarks | - $w_i$ is the latitude weighting factor used whenever spatial averaging is done to account for the spherical geometry of the Earth, with $w_i=\cos(\text{lat}_j)/\left(\sum_{i=1}^{N_{lat}}\cos(\text{lat}_j)/N_{lat}\right)$. - $\text{mean}_m$ and $\text{var}_m$ are the mean and the variance over the ensemble members respectively. | | |




# 4 Results

In this section, we evaluate the combination of four model's ensembles applying the two barycenter-based MME method described in Section 2 to the case study described in Section 3. All the results shown here are obtained for barycenters with equal weights (that is $\lambda_{\text{ECMWF}}, \lambda_{\text{NCEP}}, \lambda_{\text{ECCC}}, \lambda_{\text{KMA}} = 1/4$). We tested optimizing the weights with respect to the CRPS, but it leads to either little improvement or slight deterioration of the forecast's skill depending on the variables and scores (not shown here).

## 4.1 Validation and comparison of the barycenter-based MME

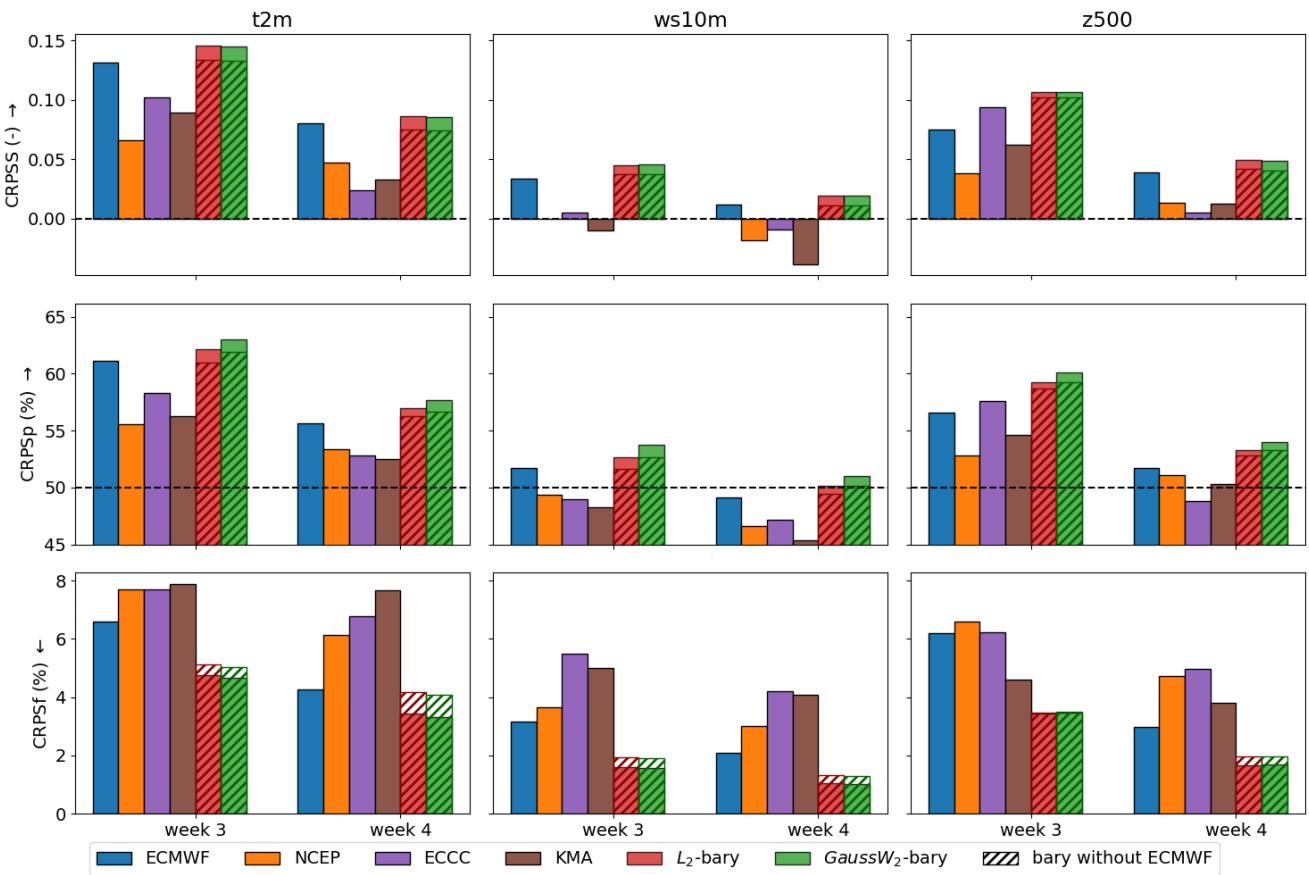

**Figure 4.** Skill scores for the weekly 2m temperature (left), 10m wind speed (center) and geopotential height at 500hPa (right): the CRPSS (top), the proportion of skillful forecasts (CRPSp, middle) and proportion of critical failure (CRPSf, bottom).

Figure 4 summarizes the performance of the four SME and the two barycenter-based MMEs. For the latter, the bars represent the performance of the barycenters combining the four SMEs (the hatched areas represent the performance of the barycenters





of three SMEs out of four ; these results are discussed below, Sect. 44.3). The distribution of the scores (CRPSS, CRPSp and

CRPSf) over the starting dates is represented by its mean in the plot, and is used to test if the model's scores are significantly different from each other at a 5% significance level. The significance is estimated using the Wilcoxon signed-rank test, a non-parametric paired statistical test (Wilcoxon, 1945). We chose to show the CRPSS here instead of the mean CRPS for an easier comparison across variables. However, the mean and the statistical tests are computed on the (spatially averaged) CRPS distributions.

For all three variables and three scores, the two barycenters outperformed the SMEs, however the gain with respect to the best performing SME is sometimes small. In particular, ECMWF has the best performance among the four SMEs for the surface variables, the 2m temperature and the 10m wind speed. In terms of CRPSS, the two barycenters have similar values and are significantly different from all the SMEs for the 10m wind speed and the geopotential height. Even though two or three of the models (depending of the week) have a negative CRPSS for the 10m wind speed, merging them with more skillful

SMEs still adds value: the barycenters have a significantly better CRPSS. For the 2m temperature, the barycenters do not have a significantly different CRPSS than ECMWF that already has a relatively large CRPSS.

In terms of CRPSp, the $GaussW_2$ barycenter is significantly better than the others, including the $L_2$ barycenter that ranks second. That is, the barycenters, and especially the $GaussW_2$ barycenter, are more often better than the climatology than the SMEs. The barycenter-based ensembles also significantly reduce the risk of big failure as shown by their lower CRPSf. The

two barycenters have similar CRPSf values, clearly lower than the values of the SMEs. In particular, the proportion of critical failures is divided by almost two for the 10m wind speed compared to the best SME, ECMWF.

Variations in scores across the spatial domain for each model are large in the sense that they tend to dominate over differences between models (not shown here). We thus expect this to be also the case for the MMEs. Figure 5 shows the differences between ECMWF and the $GaussW_2$ barycenter's scores. We chose ECMWF as reference since it has the best performance among the

SMEs for two of the three variables (the surface ones). The map of the differences between the $L_2$ barycenter and ECMWF is very similar and so is not shown here. For the CRPSS, one can see clear areas that benefit from the MMEs, but these areas are not the same depending on the variables. For the 2m temperature, the barycenter is more advantageous in the North-West, above the North Sea, Atlantic ocean and Scandinavia. ECMWF is better above Sweden and Finland for the 10m wind speed. For this variable, the barycenter benefits more central Europe as well as the South and East Europe. The spatial pattern is

smoother for the geopotential height with the largest differences above northern Europe where the barycenter is more skillful. Despite these spatial variation across the three variables, the barycenter outperforms ECMWF at a majority of grid points.

The maps for the CRPSp differences are more pixelated. However, the positive and negative areas roughly match the ones of the CRPSS differences. In contrast, the maps of the CRPSf differences show very different spatial patterns. For example, ECMWF performs better over Spain for the two surface variables for both the CRPSS and the CRPSp, but the barycenter

has a better CRPSf there. In fact, the barycenter outperforms ECMWF in terms of CRPSf almost everywhere. This means that combining ensemble forecasts has a stabilizing effect, avoiding extremely bad forecasts. However, the differences are relatively small, less than 3% at most grid points.



**Figure 5.** Maps of the differences between ECMWF and $GaussW_2$ barycenter in terms of (top) CRPSS, (middle) CRPSp, and (bottom) CRPSf averaged over weeks 3 and 4. Green indicates that $GaussW_2$ barycenter performs better and pink that ECMWF does. For readability reason, a discrete colorbar is used for the $\Delta$CRPSp and $\Delta$CRPSf which are more pixelized.

## 4.2 Ensembles calibration

In order to get insight on the calibration of the ensemble forecasts, we look at their SSR maps in Figure 6. The spatial patterns of weeks 3 and 4 being similar, only their average is displayed here. The spatial patterns are consistent across the SMEs. They tend to be under- or over-confident in the same regions but with different intensities. For example, all models are over-dispersed over Scandinavia for the 2m temperature forecasts, with KMA having the least and ECCC the most overdispersion. On the contrary, they are all under-dispersed over the Mediterranean sea.

Note that by construction the barycenters have the same ensemble mean and so the same RMSE so that the differences in their SSR are only due to their spread. Their SSRs have similar patterns as the SMES. However, one can notice that the $L_2$ barycenter tends to have more spread than the $GaussW_2$ barycenter: it shows more overdispersion and less underdispersion.





**Figure 6.** Spread-Skill-Ratio (SSR) for the weekly 2m temperature (left), 10m wind speed (center) and geopotential height at 500hPa (right) averaged over weeks 3 and 4 for the four single-model ensembles and the two barycenters. Orange indicates overdispersion (i.e. under-confident forecasts), and blue under-dispersion (i.e. overconfident forecasts).





This can be explain by the way they were constructed. The variance of the $L_2$ barycenter is indeed given by

$$\text{Var}\left(\mu_{L_2}\right) = \sum_{i=1}^{d} \lambda_i \text{Var}\left(\mu_i\right) + \sum_{i=1}^{k} \lambda_i \left(\overline{\mu_i} - \sum_{j=1}^{k} \lambda_j \overline{\mu_j}\right)^2, \tag{7}$$

where $\overline{\mu}_i$ is the mean of the distribution $\mu_i$. The variance of the barycenter is thus always equal or higher than the average

of the SME's variances. This property is advantageous in the case of under-dispersed input ensembles. However, it is an
inconvenient if they are better calibrated or already over-dispersed. On the other hand, the input ensembles are first centered
and their covariances scaled before being pooled together to build the $GaussW_2$ barycenter. Its covariance is the one of the
Gaussian barycenter (Sect. 22.22.2.2). Thus, the $GaussW_2$ barycenter is more advantageous than the $L_2$ barycenter in case of
over-dispersed input ensembles.

### 4.3  MME without ECMWF

In Section 44.1, ECMWF is found to be significantly more skillful than the other three SMEs for the two surface variables.
Moreover, although the two barycenters have overall better performances, ECMWF is still better in some regions. Thus, to in-
vestigate the importance of ECMWF within the combination, we compute the barycenters of NCEP, ECCC and KMA (without
ECMWF). They are compared to the barycenters with ECMWF in Figure 4 (hatched bar). As expected, we observe a decrease

of the performances for all scores and variables. This is also true for most grid points when looking at the differences maps
except for these points where the barycenters (with four models) are notably more skillful than ECMWF (not shown here).

When comparing the barycenters with the three SMEs used to build them, we can make similar observation as in Sec-
tion 44.1. That is, the $L_2$ and $GaussW_2$ barycenters have similar CRPSS and CRPSf, while $GaussW_2$ barycenter has a better
CRPSp (significant at the $5\%$ level) and both barycenters have better scores than the three SMEs. However, two features are

remarkable. First, the CRPSS and CRPSp differences between the barycenters and the SMEs for the surface variables are
notably larger than they were between the barycenters with four models and ECMWF. A particularly striking example is for
the CRPSS of the 10m wind speed: the three SMEs show no skill with very low or negative CRPSS, but their barycenters has
a similar CRPSS than ECMWF. Second, the barycenters with only three models are performing as well as ECMWF in terms
of CRPSS and CRPSp (there are no significant statistical differences) and better in terms of CRPSf. So, with three less skillful

models, we can reproduce the considerably better performance of ECMWF in the case studied here.

## 5  Discussion

### 5.1  $L_2$ barycenter versus $GaussW_2$ barycenter

We have seen that the MMEs improve average forecast skill compared to the SMEs with respect to all the scores and for all
the variables considered, even though a SME could perform better at some locations. In general, the two barycenters have

similar CRPSS and CRPSf, but the $GaussW_2$ barycenter has a better CRPSp. In other words, the two barycenters are similar
in average, but that the $GaussW_2$ is better more often without leading to an increase of very bad forecasts (with respect to



the CRPS). Another important difference between the barycenters is the way they represent the forecast uncertainty with their ensemble spread. The $L_2$ barycenter always has more ensemble spread than the $GaussW_2$ barycenter, which in turn has an impact on their reliability.

The two barycenters have the same means (and the latter are only dependent on the means of the input distributions), but their covariances differ. The covariances of the Wasserstein barycenter depend solely on the covariances of the ensembles (see Section 2), while the covariance of the $L_2$ barycenter depends on both their means and their variances. In fact, we can interpret the first term in the variance of the $L_2$ barycenter (Eq. (7)) as a measure of the average uncertainty of the forecast due to uncertainty in the initial condition and the second term as a measure of the model uncertainty in terms of spread of the

ensemble means. Thus, the variance of the $L_2$ barycenter accounts for model errors but ignores differences in higher moments than the ensemble mean. On the other hand, the $GaussW_2$ barycenter is merging the forecast uncertainties of the different models (i.e. the uncertainty due to the initial conditions). Instead, the information on the model uncertainty lies in the value of the barycenter distance: $\min_\mu \sum_{k=1}^{d} \lambda_k . d(\mu_k, \mu)^2$. This information is not exploited here, but it could be considered to account for model uncertainty. Finally, not only are the dynamical and the model uncertainties captured by different quantities

for the two barycenters (variance of the barycenter versus barycenter distance), but the way they are measured also differs. Understanding the implications of these differences requires further investigation.

### 5.2    Best model versus models combination

There can be several reasons why merging ECMWF with the less skillful models, is leading to barycenters with improved performances. First, the three SMEs have generally less skill than ECMWF, but can be punctually better for some given locations,

lead times or initialization dates. The barycenters can exploit this information to improve skill thanks to error cancellation and to the non-linearity of the skill metrics (Hagedorn et al., 2005). Second, the ECMWF forecasts may have good performances but be overconfident. In that case, adding other models with lower skills increases the spread of the ensemble and can move the ensemble mean towards the truth, as shown by Weigel et al. (2008) for seasonal ensemble forecasts. For the $L_2$ barycenter, this can be seen from Equation (6) which shows that the CRPS of the $L_2$ barycenter is composed of two parts: the weighted

average of the CRPS of the SMEs and minus the weighted average of the CRPS between all the different pairs of SMEs. Thus, if one model has a worse CRPS than an other, it can still improve the CRPS of the barycenter if the CRPS between the models is large enough to compensate its own CRPS.

### 5.3    Weighted barycenters

Equal weighting of the models in the pooling method is the simpler and most used approach. However, some studies investigate

the use weighted multi-model ensembles with divergent results (at weather or seasonal scale in Weigel et al. (2008); Casanova and Ahrens (2009); Kharin and Zwiers (2002) and at climate scale in Haughton et al. (2015)). At the subseasonal scale, Wanders and Wood (2016) used muti-variable linear regression on the ensemble means to derive the model weights. They show that the weighted muti-model ensemble have a better deterministic performance but also probabilistic one (in terms of Brier score) than the non-weighted multi-model.





Here, we have shown results giving equal weights to the models, but the barycenter formulation easily allows for the construction of weighted-MMEs. Results for weights that do not depend on space or time minimizing the mean CRPS were also obtained but not shown because, contrary to Wanders and Wood (2016), no significant improvement was found. One explanation could be the limited time period used here. The collection of forecasts by the S2S database started in 2015. We thus have less than ten years of data, which may not be enough to find stable weights, as shown by Wanders and Wood (2016); Kharin and

Zwiers (2002). Using the reforecasts (instead of the forecasts) would allow one to extend the time period. However, it would still be limited to twelve years (due to NCEP limited reforecast period) and would lead to additional difficulties. In particular, the starting dates of the reforecasts for the different models in the S2S database do not match. In addition, the reforecasts have fewer members than the forecasts so that the reforecasts results may not be transferred to the forecasts without additional assumptions.

Yet, another strategy could be to allow for the weights to depend on space. We indeed saw that even if the barycenters tend to be more skillful than the SMEs, they can be outperformed by ECMWF for some scores at some locations. This suggests that giving more weight to ECMWF in these regions would be beneficial. Similarly, even if the barycenter without ECMWF showed a general degradation of the scores compared to the barycenters with the four models, it did perform better in some specific regions. Thus the barycenter with four models would benefit from giving less or no weight to ECMWF in these regions.

However, optimizing weights grid point by grid point requires more data and may lead to spatial inconsistencies (Wanders and Wood, 2016; DelSole et al., 2013). An alternative, would be to learn weights per region (such as in Wanders and Wood (2016)) or enforcing some spatial smoothness on the weights (also acting as a regularization and helping with over-fitting).

## 6   Conclusions

We explore methods to combine ensemble forecasts from multiple models based on barycenters of forecast ensembles. Building
on the recognition of the relevance of probabilistic forecasts for S2S prediction, we work directly in the probability distribution space. That is, the ensemble forecasts are manipulated as discrete probability distributions. This allows us to use existing tools from this space and in particular the notion of barycenter. Here, we explore two barycenters based on different metrics: the $L_2$ distance and the Wasserstein ($GaussW_2$)-distance. We show that the $L_2$ barycenter is in fact equivalent to the well-known pooling method and compare it to the new $GaussW_2$ barycenter based method. Moreover, the variance of the $L_2$ barycenter
can be decomposed into a term representing the average uncertainty of individual forecasts and a term representing model error, while the variance of the $GaussW_2$ barycenter does not account for the latter. It is instead the $GaussW_2$ barycenter value that does.

    This first application of our framework to S2S prediction is illustrated through the combination of four single-model ensembles to predict the winter surface temperature, surface wind speed and 500hPa geopotential height over Europe. By construction,
the two barycenters have the same ensemble means, meaning they have the same performance in terms of deterministic scores. However, they differ by the way they represent initial conditions and model uncertainty. In particular, by construction the $L_2$ barycenter is likely to be under-confident due to overdispersion, implying that the $L_2$ barycenter is advantageous in the case





of under-dispersive input ensembles, but disadvantageous in terms of calibration otherwise. More generally, we show that both
barycenters perform similarly on average but that the $GaussW_2$ barycenter is better more often (across the different grid points
and dates) with respect to the CRPS. Even in cases where one of the SMEs has superior skill compared to the others (ECMWF
for the surface variables), it is still advantageous to combine them into a barycenter. This confirms the interest of multi-model
methods shown by previous studies. We also show that by using the other three models, one can build barycenter-based MMEs
that are as skillful as the best SME.

This study is a proof of concept to develop the framework and investigate the properties of the barycenter-based MMEs.
These results constitute a promising first step towards improving S2S predictions using barycenters to merge ensemble fore-
casts. A next step would be to further investigate the optimization of the weights in the barycenter in order to build weighted
MMEs. Another interesting question concerns the possibility of building multivariate Wasserstein barycenters. The $L_2$ barycen-
ter does not take into account the covariances between the variables, whereas the Wasserstein barycenter does. Finally, the im-
plications of the fact that both barycenters account differently for forecast and model uncertainty deserve further investigation.

*Code and data availability.* The Python code for the barycenter-based MMEs is openly available on Github (https://github.com/clecoz/OT_
for_MME) and Zenodo (https://zenodo.org/records/15058503, Le Coz et al. (2025a)).

This work is based on S2S data. S2S is a joint initiative of the World Weather Research Programme (WWRP) and the World Climate
Research Programme (WCRP). The original S2S database is hosted at ECMWF as an extension of the TIGGE database (https://apps.
ecmwf.int/datasets/data/s2s/). MERRA-2 data is available from the Goddard Earth Sciences Data and Information Services Center (GES
DISC) data archive (see Global Modeling and Assimilation Office (GMAO), 2015a, b, c). The pre-processed data are shared on Zenodo
(https://doi.org/10.5281/zenodo.15038871, Le Coz et al. (2025b)).

## Appendix A: Variance shrinkage in discrete Wasserstein barycenter

A potential problem of discrete Wasserstein barycenter is its reduced variance. Indeed, computing the Wasserstein barycen-
ter directly on discrete probability distributions leads to a variance shrinkage problem which worsen with increasing space
dimension and decreasing distribution sampling. This problem can be illustrated with the following empirical experiment:

1. Create two discrete distributions by randomly drawing $n$ samples from two normal distributions with unit-covariance in
   $\mathbb{R}^d$.

2. Compute the Wasserstein barycenter of the two discrete distributions (using the closed-form expression here).

3. Compute the normalized trace of the covariance of the barycenter divided by the number of variables (that is, the average
variance of the variables).

4. Repeat 100 times.





We repeated this experiment for different sampling sizes $n$ and space dimensions $d$. The results of this experiment are shown in Figure A1. By construction, the Wasserstein barycenter of the two normal distributions with unit-covariance also has a unit-variance (and so an average variance of one). However, one can observe that the discrete barycenter has a lower average variance. The average variance shrinks as the sample size decreases and the dimension increases. For a number of dimensions equal to six and 50 samples (similar to the setting of our application), we observe almost $15\%$ loss of the average variance.

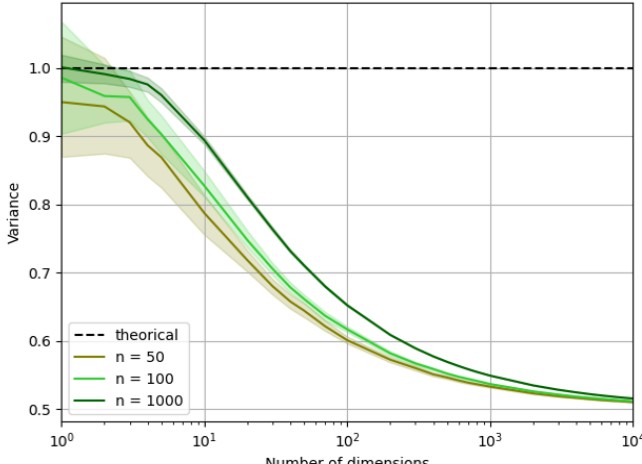

**Figure A1.** Average variance of the discrete Wasserstein barycenter as a function of the dimension $d$ for different sample size $n$. The full line represents this variance averaged over the 100 iterations with the minimum and maximum values indicated by the envelop.

## Appendix B: Energy distance and its associated barycenter

Let $\mu_1$ and $\mu_2$ be two distributions, and $F_1$ and $F_2$ be their CDF. That is, $F_1(x) = \int_{-\infty}^{x} \mu_1(t)dt$ and $F_2(x) = \int_{-\infty}^{x} \mu_2(t)dt$ $\forall x \in \mathbb{R}$. The squared energy distance between $\mu_1$ and $\mu_2$ is

$$\mathcal{E}^2(\mu_1, \mu_2) = \int_{-\infty}^{+\infty} [F_1(x) - F_2(x)]^2 \, dx = \int_{-\infty}^{+\infty} \left[ \int_{-\infty}^{x} \mu_1(t)dt - \int_{-\infty}^{x} \mu_2(t)dt \right]^2 dx$$





The energy barycenter of $\mu_1$ and $\mu_2$ is the solution of the following minimization problem $\mu_{\mathcal{E}}^{\alpha} = \arg\min_{\mu} \alpha \mathcal{E}^2(\mu, \mu_1) + (1 - \alpha)\mathcal{E}^2(\mu, \mu_2)$. Let $\mathcal{B}(\mu) = \alpha \mathcal{E}^2(\mu, \mu_1) + (1 - \alpha)\mathcal{E}^2(\mu, \mu_2)$, then we have:

$$
\begin{aligned}
\frac{d}{d\mu}\mathcal{B}(\mu) &= \frac{d}{d\mu}\left[ \alpha \int_{-\infty}^{+\infty} [F(x) - F_1(x)]^2\, dx + (1 - \alpha) \int_{-\infty}^{+\infty} [F(x) - F_2(x)]^2\, dx \right] \\
&= \frac{d}{d\mu} \int_{-\infty}^{+\infty} \left( \alpha \left[ \int_{-\infty}^{x} \mu(t)dt - \int_{-\infty}^{x} \mu_1(t)dt \right]^2 + (1 - \alpha)\left[ \int_{-\infty}^{x} \mu(t)dt - \int_{-\infty}^{x} \mu_2(t)dt \right]^2 \right) dx \\
&= \int_{-\infty}^{+\infty} \frac{d}{d\mu}\left( \alpha \left[ \int_{-\infty}^{x} (\mu(t) - \mu_1(t))\, dt \right]^2 + (1 - \alpha)\left[ \int_{-\infty}^{x} (\mu(t) - \mu_2(t))\, dt \right]^2 \right) dx \\
&= \int_{-\infty}^{+\infty} \left( 2\alpha \int_{-\infty}^{x} [\mu(t) - \mu_1(t)]\, dt\, \mu(x) \right) dx + \int_{-\infty}^{+\infty} \left( 2(1 - \alpha) \int_{-\infty}^{x} [\mu(t) - \mu_2(t)]\, dt\, \mu(x) \right) dx \\
&= 2 \int_{-\infty}^{+\infty} \left( \int_{-\infty}^{x} (\mu(t) - [\alpha \mu_1(t) + (1 - \alpha)\mu_2(t)])\, dt \right) \mu(x)dx
\end{aligned}
$$

Thus,

$$
\frac{d}{d\mu}\mathcal{B}(\mu_{L_2}^{\alpha}) = 0
$$

where $\mu_{L_2}^{\alpha} = \alpha \mu_1 + (1 - \alpha)\mu_2$ is the $L_2$ barycenter. The $L_2$ barycenter is also a barycenter for the energy distance in 1D.

## Appendix C: CRPS and $L_2$-barycenter

Let $\mu_1, ..., \mu_d$ be $d$ probability distributions, and $F_1, ..., F_d$ their respective cumulative density functions. Their weighted-L2 barycenter is the probability distribution $\mu_{L_2}^{\lambda}$ given by

$$
\mu_{L_2}^{\lambda} = \sum_{i=1}^{n} \lambda_i \mu_i \tag{C1}
$$

where $\lambda = (a_i, ...\lambda_n)$ are the barycentric weights such that $\sum_{i=1}^{n} \lambda_i = 1$.

Let $\nu_{obs}$ be the probability distribution of the truth or reference, and $F_{obs}$ its cumulative density function. In the case of a deterministic observation $y_r$, $\nu_{obs} = \delta_{y_{obs}}$ is a Dirac and $F_{obs}$ a step function.

The CRPS of two probability distributions is defined as the squared $L_2$-distance between their CDFs:

$$
\mathrm{CRPS}(\mu_i, \mu_j) = \int_{-\infty}^{+\infty} [F_i(u) - F_j(u)]^2\, du \tag{C2}
$$



Thus, we have

$$\text{CRPS}(\mu_{L_2}^\lambda, \nu_{obs}) = \int_{-\infty}^{+\infty} \left[ F_{L_2}^\lambda(u) - F_{obs}(u) \right]^2 du$$

$$= \int_{-\infty}^{+\infty} \left[ \sum_{i=1}^n \lambda_i F_i(u) - F_{obs}(u) \right]^2 du$$

$$= \int_{-\infty}^{+\infty} \left[ \sum_{i=1}^n \lambda_i^2 F_i(u)^2 + F_{obs}(u)^2 + \sum_{i=1}^n \sum_{j \neq i} \lambda_i \lambda_j F_i(u) F_j(u) - 2 \sum_{i=1}^n \lambda_i F_i(u) F_{obs}(u) \right] du$$

knowing that $\sum_{i=1}^n \lambda_i = 1 \Rightarrow \lambda_i = 1 - \sum_{j \neq i} \lambda_j \Rightarrow \lambda_i^2 = \lambda_i(1 - \sum_{j \neq i} \lambda_j) = \lambda_i - \lambda_i \sum_{j \neq i} \lambda_j$

$$= \int_{-\infty}^{+\infty} \left[ \sum_{i=1}^n \lambda_i F_i(u)^2 + \sum_{i=1}^n \sum_{j \neq i} \lambda_i \lambda_j F_i(u)^2 + \sum_{i=1}^n \lambda_i F_{obs}(u)^2 \right.$$

$$\left. + 2 \sum_{i=1}^n \sum_{j \geq i} \lambda_i \lambda_j F_i(u) F_j(u) - 2 \sum_{i=1}^n \lambda_i F_i(u) F_{obs}(u) \right] du$$

$$= \int_{-\infty}^{+\infty} \left[ \sum_{i=1}^n \lambda_i [F_i(u)^2 + F_{obs}(u)^2 - 2 F_i(u) F_{obs}(u)] \right.$$

$$\left. - \sum_{i=1}^n \sum_{j \geq i} \lambda_i \lambda_j [F_i(u)^2 + F_j(u)^2 - 2 F_i(u) F_j(u)] \right] du$$

$$= \sum_{i=1}^n \lambda_i \left( \int_{-\infty}^{+\infty} [F_i(u) - F_{obs}(u)]^2 du \right) - \sum_{i=1}^n \sum_{j \geq i} \lambda_i \lambda_j \left( \int_{-\infty}^{+\infty} [F_i(u) - F_j(u)]^2 du \right)$$

$$= \sum_{i=1}^n \lambda_i \text{CRPS}(\mu_i, \nu_{obs}) - \sum_{i=1}^n \sum_{j \geq i} \lambda_i \lambda_j \text{CRPS}(\mu_i, \mu_j)$$

$$\text{CRPS}(\mu_{L_2}^\lambda, \nu_{obs}) = \sum_{i=1}^n \lambda_i \text{CRPS}(\mu_i, \nu_{obs}) - \sum_{i=1}^n \sum_{j \neq i} \frac{1}{2} \lambda_i \lambda_j \text{CRPS}(\mu_i, \mu_j) \tag{C3}$$

## Appendix D: Spatial performance's significance

### D1 Barycenters with four models





**Table D1.** Two-sided Wilcoxon test's p-value for the 2m temperature (t2m) for weeks 3 and 4 combined. Bold indicates that the models are significantly different at the 5% significance level (i.e. p-values< 0.05).

|  |  | ECMWF | NCEP | ECCC | KMA | $L_2$-bary | $GaussW_2$-bary |
|---|---|---|---|---|---|---|---|
| CRPSmean | $L_2$-bary | 3.5e-01 | **3.5e-09** | **3.3e-09** | **5.8e-09** | - | 3.6e-01 |
|  | $GaussW_2$-bary | 3.4e-01 | **2.6e-09** | **2.8e-09** | **1.1e-08** | 3.6e-01 | - |
| CRPSp | $L_2$-bary | 9.0e-02 | **4.4e-10** | **2.0e-09** | **1.7e-12** | - | **6.5e-16** |
|  | $GaussW_2$-bary | **2.5e-03** | **2.3e-12** | **5.9e-12** | **7.1e-15** | **6.5e-16** | - |
| CRPSf | $L_2$-bary | **1.1e-04** | **6.9e-17** | **1.8e-20** | **2.0e-17** | - | **3.9e-05** |
|  | $GaussW_2$-bary | **2.1e-05** | **3.4e-18** | **8.9e-21** | **5.9e-18** | **3.9e-05** | - |

**Table D2.** Two-sided Wilcoxon test's p-value for the 10m wind speed (ws10m) for weeks 3 and 4 combined. Bold indicates that the models are significantly different at the 5% significance level (i.e. p-values< 0.05).

|  |  | ECMWF | NCEP | ECCC | KMA | $L_2$-bary | $GaussW_2$-bary |
|---|---|---|---|---|---|---|---|
| CRPSmean | $L_2$-bary | **2.5e-02** | **1.2e-16** | **6.8e-12** | **6.7e-18** | - | 7.2e-01 |
|  | $GaussW_2$-bary | **2.0e-02** | **2.3e-16** | **1.1e-11** | **5.5e-18** | 7.2e-01 | - |
| CRPSp | $L_2$-bary | **1.5e-02** | **6.3e-12** | **1.7e-11** | **1.5e-14** | - | **1.1e-22** |
|  | $GaussW_2$-bary | **1.3e-05** | **9.7e-17** | **3.6e-16** | **1.2e-18** | **1.1e-22** | - |
| CRPSf | $L_2$-bary | **1.4e-19** | **1.4e-26** | **6.3e-31** | **1.2e-26** | - | **2.1e-03** |
|  | $GaussW_2$-bary | **2.3e-20** | **1.3e-26** | **6.0e-31** | **8.4e-27** | **2.1e-03** | - |

**Table D3.** Two-sided Wilcoxon test's p-value for the geopotential height (z500) for weeks 3 and 4 combined. Bold indicates that the models are significantly different at the 5% significance level (i.e. p-values< 0.05).

|  |  | ECMWF | NCEP | ECCC | KMA | $L_2$-bary | $GaussW_2$-bary |
|---|---|---|---|---|---|---|---|
| CRPSmean | $L_2$-bary | **9.4e-03** | **8.1e-08** | **4.8e-04** | **3.7e-06** | - | 3.0e-01 |
|  | $GaussW_2$-bary | **1.0e-02** | **7.2e-08** | **4.3e-04** | **3.2e-06** | 3.0e-01 | - |
| CRPSp | $L_2$-bary | **1.3e-03** | **3.6e-06** | **8.4e-05** | **1.1e-05** | - | **2.4e-12** |
|  | $GaussW_2$-bary | **3.0e-05** | **9.4e-08** | **1.6e-06** | **1.2e-07** | **2.4e-12** | - |
| CRPSf | $L_2$-bary | **2.6e-09** | **2.0e-13** | **2.5e-17** | **6.2e-08** | - | 9.8e-01 |
|  | $GaussW_2$-bary | **2.1e-09** | **1.9e-13** | **5.7e-17** | **6.5e-08** | 9.8e-01 | - |





## D2   Barycenters with three models

**Table D4.** Two-sided Wilcoxon test's p-value for the 2m temperature (t2m) for weeks 3 and 4 combined. Bold indicates that the models are significantly different at the $5\%$ significance level (i.e. p-values$< 0.05$).

| | | ECMWF | NCEP | ECCC | KMA | $L_2$-bary 4 models | $L_2$-bary 3 models | $GaussW_2$-bary 4 models | $GaussW_2$-bary 3 models |
|---|---|---|---|---|---|---|---|---|---|
| CRPSmean | $L_2$-bary (3 models) | 6.4e-01 | **1.4e-07** | **4.4e-06** | **3.3e-07** | **4.1e-06** | - | **5.6e-06** | 1.8e-01 |
| | $GaussW_2$-bary (3 models) | 5.8e-01 | **1.5e-07** | **5.7e-06** | **4.5e-07** | **6.8e-06** | 1.8e-01 | **2.7e-06** | - |
| CRPSp | $L_2$-bary (3 models) | 8.9e-01 | **3.3e-08** | **1.4e-06** | **2.0e-10** | **3.6e-06** | - | **5.4e-12** | **3.4e-10** |
| | $GaussW_2$-bary (3 models) | 2.7e-01 | **9.3e-11** | **1.5e-08** | **1.8e-12** | 2.1e-01 | **3.4e-10** | **8.3e-07** | - |
| CRPSf | $L_2$-bary (3 models) | 1.2e-01 | **6.8e-13** | **5.4e-17** | **7.2e-16** | **1.5e-07** | - | **6.7e-10** | **4.1e-02** |
| | $GaussW_2$-bary (3 models) | 7.6e-02 | **6.4e-14** | **5.3e-17** | **2.7e-16** | **9.7e-06** | **4.1e-02** | **2.5e-08** | - |



**Table D5.** Two-sided Wilcoxon test's p-value for the 10m wind speed (ws10m) for weeks 3 and 4 combined. Bold indicates that the models are significantly different at the $5\%$ significance level (i.e. p-values$< 0.05$).

| | | ECMWF | NCEP | ECCC | KMA | $L_2$-bary 4 models | $L_2$-bary 3 models | $GaussW_2$-bary 4 models | $GaussW_2$-bary 3 models |
|---|---|---|---|---|---|---|---|---|---|
| CRPSmean | $L_2$-bary (3 models) | 9.9e-01 | **4.9e-13** | **7.7e-09** | **3.3e-15** | **9.1e-10** | - | **4.6e-10** | 4.8e-01 |
| | $GaussW_2$-bary (3 models) | 8.2e-01 | **5.9e-13** | **9.2e-09** | **2.0e-15** | **2.1e-07** | 4.8e-01 | **2.9e-09** | - |
| CRPSp | $L_2$-bary (3 models) | 6.1e-01 | **3.1e-09** | **5.8e-08** | **1.3e-12** | **3.1e-06** | - | **9.7e-17** | **4.5e-20** |
| | $GaussW_2$-bary (3 models) | 5.1e-02 | **6.4e-14** | **1.6e-11** | **2.6e-17** | 8.9e-01 | **4.5e-20** | **1.1e-06** | - |
| CRPSf | $L_2$-bary (3 models) | **3.3e-09** | **6.9e-22** | **8.8e-30** | **1.4e-25** | **2.3e-11** | - | **4.2e-13** | **3.8e-02** |
| | $GaussW_2$-bary (3 models) | **4.1e-10** | **4.3e-22** | **6.2e-30** | **7.7e-26** | **1.7e-08** | **3.8e-02** | **2.1e-11** | - |

**Table D6.** Two-sided Wilcoxon test's p-value for the geopotential height (z500) for weeks 3 and 4 combined. Bold indicates that the models are significantly different at the $5\%$ significance level (i.e. p-values$< 0.05$).

| | | ECMWF | NCEP | ECCC | KMA | $L_2$-bary 4 models | $L_2$-bary 3 models | $GaussW_2$-bary 4 models | $GaussW_2$-bary 3 models |
|---|---|---|---|---|---|---|---|---|---|
| CRPSmean | $L_2$-bary (3 models) | 2.0e-01 | **7.9e-07** | **3.3e-03** | **2.1e-05** | **1.3e-02** | - | **1.0e-02** | 2.4e-01 |
| | $GaussW_2$-bary (3 models) | 1.9e-01 | **8.9e-07** | **2.8e-03** | **2.4e-05** | **2.0e-02** | 2.4e-01 | **9.2e-03** | - |
| CRPSp | $L_2$-bary (3 models) | 7.2e-02 | **4.9e-06** | **6.5e-04** | **8.8e-05** | 7.9e-02 | - | **5.9e-05** | **2.8e-07** |
| | $GaussW_2$-bary (3 models) | **1.8e-02** | **3.7e-07** | **4.3e-05** | **2.9e-06** | 5.7e-01 | **2.8e-07** | **3.1e-02** | - |
| CRPSf | $L_2$-bary (3 models) | **4.3e-05** | **9.2e-13** | **1.0e-13** | **8.6e-06** | **5.4e-03** | - | **2.3e-03** | 8.3e-01 |
| | $GaussW_2$-bary (3 models) | **2.5e-05** | **5.9e-13** | **5.5e-14** | **9.2e-06** | **1.3e-02** | 8.3e-01 | **3.2e-03** | - |



*Author contributions.* Conceptualization: CLC, AT, RF and RP; funding acquisition: AT and RF; formal analysis: CLC; methodology: CLC, AT, RF and RP; writing - original draft report: CLC; writing - review and editing: AT, RF and RP.

*Competing interests.* The authors declare that they have no conflict of interest.

*Acknowledgements.* The authors thank Naveen Goutham for code and advice on forecast calibration. The authors thank the institut de Mathématiques pour la Planète Terre for (partial) funding (iMPT-AAP2021). This work was partially supported by the grant ANR-23-ERCC-0006 from Agence nationale de la recherche (ANR). This research was produced within the framework of Energy4Climate Interdisciplinary Center (E4C) of IP Paris and Ecole des Ponts ParisTech. This research was supported by 3rd Programme d'Investissements d'Avenir [ANR-18-EUR-0006-02].



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
