# Peer review of "A barycenter-based approach for the multi-model ensembling of subseasonal forecasts"

_EGUsphere, 2025_

## Author Comment (AC1)

We would like to thank anonymous reviewer #1 for their careful review and insightful comments. Please find below our point-by-point responses.

**General comments:**

The article explores a new method to aggregate members from multimodel ensemble, based on an optimization of the Wasserstein distance. It is applied to univariate point forecast timeseries, on a few weather parameters. It is compared with the more usual (and simpler) member pooling technique, also called the L2 barycenter, using a long range forecast database. The scores show clear benefits of combining several ensembles, which was already known for the pooling method, and they suggest that the Wasserstein-based barycenter may be slightly better than the L2 one according to some performance measures. The evaluation shows that the Wasserstein method decreases a lot the ensemble dispersion, which is a direct consequence of its design.

Is it a good thing to reduce intermodel ensemble spread in a multimodel ensemble ? One could argue that it defeats the purpose of using multiple models. Many weather prediction centers actually use multiple models to increase ensemble dispersion, because it improves key aspects of the forecasts.

Mathematically, the need to reduce spread depends on the presence of over- or under-dispersion in the raw (i.e., pooled) ensemble. In the article setup, the usefulness of implementing the Wasserstein barycenter method is not very clear, as the advertised score benefits could probably have been obtained by (technically and conceptually) simpler methods to reduce the ensemble dispersion, like EMOS or a member-by-member bias correction.

The need to reduce/increase spread depends indeed on the over/under-estimation of the raw ensemble. The GaussW2 barycenter was not used here as a way to reduce the spread but to explore a different way to combine the forecast uncertainty from different models.

An advantage of the Wasserstein barycenter compared to EMOS is that it is a non-parametric method. The Gaussian hypothesis is only used to derive the mappings, not for the final distribution/ensemble.

One understands that the Wasserstein barycenter method can get quite complex and expensive if the workspace dimension is increased (here it is only 6, owing to the time averaging applied to the forecasts), or if the distributions are significantly non-gaussian, which will restrict its practical applicability.

We agree that the number of dimensions is a bottleneck of the Wasserstein barycenter in terms of computational cost. While our study uses four dimensions, higher dimensions are feasible at reasonable cost, which scales roughly cubically with the number of dimensions. The main requirement is to estimate the covariance matrix accurately, which requires more samples for higher dimensions but regularization techniques can help mitigate this limitation.

It is therefore important to carefully select the variables to include. Here, we use the same parameter (2m temperature, 10m wind speed or geopotential height) at different time steps, but one could also consider different (correlated) parameters at the same time step. Also there exist efficient estimators for Gaussian mapping on large dimensions (especially in time) relying on diagonalization of the covariance matrix with FFT (Gnassounou 2023).

Regarding the non-Gaussian distributions, the Gaussian assumption is used here only to derive the mappings, which are then applied to the original (discrete) distributions without assuming a specific shape. Non-Gaussianity was not an issue in our case (including for the wind speed), and was not explored further here. For strongly non-Gaussian variables, one could apply a transformation to make the distribution more Gaussian (for example Gaussian anamorphosis, see Bertino et al. 2003 and Lussana et al. 2021), compute the barycenter in the transformed space, and then apply the inverse transform.

References:

Bertino, L., Evensen, G. and Wackernagel, H. (2003), Sequential Data Assimilation Techniques in Oceanography. International Statistical Review, 71: 223-241. https://doi.org/10.1111/j.1751-5823.2003.tb00194.x

Lussana, C., Nipen, T. N., Seierstad, I. A., and Elo, C. A.: Ensemble-based statistical interpolation with Gaussian anamorphosis for the spatial analysis of precipitation, Nonlin. Processes Geophys., 28, 61–91, https://doi.org/10.5194/npg-28-61-2021, 2021.

Gnassounou, T., Flamary, R., & Gramfort, A. (2023). Convolution Monge mapping normalization for learning on sleep data. Advances in Neural Information Processing Systems, 36, 10457-10476.

Despite these rather underwhelming results, the article can serve as a nicely written course on the Wasserstein distance, which is well known in the field of AI (e.g. as a probabilistic metric), but not yet in meteorology, although similar spatial verification concepts (such as SAL) are standard.

The idea of using the Wasserstein metric as an alternative to CRPS for comparing ensemble forecast distributions is original, it might be useful for other applications than the aggregation of multimodel ensembles (perhaps for clustering ?).

In a nutshell, the article does not present a clearly useful application, but it is a thought-provoking introduction to a so far little used tool that might lead to practical uses in the future.

Thank you for your positive feedback, we indeed do not claim that Gaussian Wasserstein barycenters are always a better solution but think that it is a nice and original tool that the community could benefit from.

**Specific remarks (tagged by manuscript line number)**

Some parts of the text are unclear or ambiguous. There are typos.

In some languages, the term "barycenter" is often used to mean "centroid" (i.e., weight averages), which can be confusing because centroid extraction is commonly used in ensemble post-processing to summarize ensembles as pseudo-deterministic forecast scenarios. It would help to clarify early in the introduction that the aim is to transform an ensemble distribution into another distribution, not into a single value.

l.19, 18: replace "single-model" by "SME" or equivalent.

The "single-model" will be replaced by SME as suggested.

l.29: insert "which _often_ implies" (a poor MME setup can actually degrade the forecast PDFs)

Thank you for your comment. This will be implemented.

l. 62-63: do you mean "in the BMA method, _no_ assumtion is made" (otherwise it contradicts the previous sentence on EMOS)

The EMOS method assumes a specific parametric model for the shape of the output distribution, while the BMA method assumes a parametric model for the shape of the input distributions. This passage will be reformulated to be more precise and remove the confusion.

*"Other approaches, such as the Ensemble Model Output Statistics (EMOS) and Bayesian Model Averaging (BMA), adopt a probabilistic perspective and aim to construct one probability density function (PDF) from the different ensemble forecasts. The EMOS method assumes a specific parametric form for this predictive PDF, the parameters being obtained from the input ensembles by optimizing with respect to a chosen score (e.g.~the CRPS) over a training period (Gneiting et al, 2005). In contrast, the BMA assumes separate parametric forms for the PDF of the input models (based on the ensemble forecasts). The predictive PDF is then a weighted mixture of these input PDFs, with weights given by the posterior probabilities of the input ensembles (Raftery et al, 2005)."*

l. 69: at this point you may clarify that the barycenter is a transformed ensemble prediction.

Thank you for pointing out this potential source of confusion. As you suggested, we will clarify by adding the following sentence "*The barycenter then serves as a tool for merging several ensemble distributions into a single transformed ensemble distribution*".

l. 136: pooling was not actually introduced because it minimizes L2, but because it is the correct thing to do if one assumes all members to be independent, identically distributed (iid) samples of the same distribution. Thus, using another method implies that one assumes the distributions sampled by each ensemble are not the same (presumably because they contain model-specific errors, which is a reasonable hypothesis according to previous

literature). This should be more explicitly stated. Based on this article I would conclude that the Wasserstein barycenter is a statistical device to represent model-specific error.

Thank you for this very good point. We will make that clearer in the text as follows: "*The pooling method is based on the assumption that all members from all ensembles are independent and identically distributed samples of the same distribution. Under this assumption, the members can be pooled to form an empirical distribution. However, this hypothesis is rarely verified in practice (Kioutsioukis and Galmarini, 2014;Knutti et al., 2017). If we assume instead that the input ensembles have different distributions because of model-specific errors (which we aim to sample), then interpreting the "pooling" method as a L2 barycenter allows us to extend it to different distances. In the following, we introduce another distance, the Wasserstein distance, and its associated barycenter. From this point onward, we consistently use "L2 barycenter" to refer to "pooling" in order to facilitate comparison with this second barycenter-based MME method*"

l.156: what do you mean by "space dimension" in appendix A ? Is it dimension d ? Or the geographical geographical spatial distribution that you discuss in Fig 5 ? How do you reach a value of 6 (in l. 495; this only explained later) ? Please replace "space" by a more precise term. According to l.104, it should be something like "the number of lead times".

Thank you for pointing out this imprecision. "Space dimension" will be replaced by "number of dimensions" in both line 156 and the appendix.

L156 will become "*This variance shrinkage, illustrated in  see appendix A, is due to the limited sample size of these distributions with respect to the number of dimensions (corresponding here to the number of lead times, see Section 2.1)*".

In Appendix A, we will also add an explanation for the values of 4 and 50 (6 was a typo): "*For four dimensions and 50 samples, we observe a loss of more than 10% of the average variance. This setting is comparable to our application, where the number of dimensionsd corresponds to the number of lead times (i.e. four weeks), and the number of samplesn corresponds to the ensemble size (ranging from 21 to 51 members, see Section 3.1.1).*"

l.162: please correct "the statistical the difficulty" and fix the Flamary reference.

The typo will be fixed ("*the statistical difficulty*)" and the parenthesis will be added for the Flamary reference.

l.165: in this context, "multivariate" could also mean "joint distribution between several meteorological variables". Please fix.

Thanks for pointing out this ambiguity. To clarify we will add the following "*Note: in this study, the multivariate distribution does not represent the join distribution between several meteorological variables, but the join distribution between different time steps of the same meteorological variables (see Section 2.1)*"

l.182: duplicate "hypothesis"

One of the "hypothesis" will be removed.

l.183: most weather variables of interest, like wind speed or precipitation, have strongly non-gaussian distributions, so that using mappings to displace their gaussian approximations will change their dispersion. These variables have a nonlinear relationship between their average and dispersion, which is why they are usually modelled as e.g. Weibull or gamma distributions.

Thank you for pointing out this gap. For variables with highly non-Gaussian distribution ,it is possible to first pre-process the data to obtain a more Gaussian distribution. Many statistical and data assimilation methods rely on a Gaussian assumption, which is not true for many variables of interest. It is thus a rather common approach to pre-process/standardize the data.

Also note that the Gaussian Assumption is used only to compute a barycenter and mapping for each model (hence focusing on first and second order moments only)  and that all non-Gaussianity of the individual samples will be preserved in the final distribution.

This will be added in the paragraph  "*For variables with a strongly non-Gaussian distribution (such as precipitation), it is also possible to first apply a transformation to make the distribution more Gaussian (e.g., using Gaussian anamorphosis; see Bertinao et al, 2003; Lussana et al, 2021), compute the barycenter in the transformed space, and then apply the inverse transform*".

l.189: again, replace "dimension of the sample space" which is confusing, by "the number of lead times".

L189 will become "*and d the number of dimensions, here d=nt lead times*" to be more precise.

l.192: define "OT"

"OT" will be replaced by "optimal transport".

Fig.2 and l.204: please use consistent terminology, e.g. "pooling" instead of "L2-barycenter" or "concatenation", since they are the same thing.

In the introduction, we use the term "pooling" to be inline with the literature. In Section 2.2.1, we show the equivalence between pooling and L2-barycenter. After that point, we chose to consistently use the term "L2 barycenter" in order to draw a parallel with the GaussW2 barycenter and facilitate comparison. We will make it clearer in Section 2.2.1 by stating clearly "*From this point, we consistently use L2 barycenter to refer to pooling in order to facilitate comparison with this second barycenter-based MME method*".

l.210-214: this discussion should give some physical reasons why preserving distribution shape can be regarded as desirable or not. Why use a multimodel ensemble in the first

place if differences in distributions are regarded as a problem to fix ? There seems to be an implicit assumption that these differences stem from climatological differences between the single-model ensembles used, so that the motivation for using the GaussW2 barycenter is to implement a bias correction scheme.

We thank you for this insightful comment. Our motivation to use the GaussW2 barycenter was not to fix the differences between input distributions or preserve a specific distribution shape. Rather, we found it interesting to explore how ensemble forecasts can be manipulated in the space of probability distribution using different tools. The Wasserstein distance has very different properties than the L2 distance.

We also acknowledge that bimodality of the PDF of a forecast variable can be of interest and have implications (Bertossa et al. 2021). In situations where different models would explore very different outcomes, some bimodality could emerge from a multi-model ensemble using the L2 barycenter, whereas the GaussW2 barycenter is likely to remove this bimodality. In a single model ensemble, we have found bimodality to be rather uncommon and confined to specific regions (above the sharp temperature gradients of Western Boundary Currents for instance, Bertossa et al. 2021). It would remain to be explored, how much more frequently bimodality emerges from a multi-model ensemble.

Reference:

Bertossa, C., Hitchcock, P., DeGaetano, A., and Plougonven, R.: Bimodality in ensemble forecasts of 2 m temperature: identification, Weather Clim. Dynam., 2, 1209–1224, https://doi.org/10.5194/wcd-2-1209-2021, 2021.

 l.284: replace "simulation's number" by "ensemble member index"

It will be replaced.

 l.288: compare -> compares

The typo was corrected.

l.294: a drawback of the CRPSf definition is that it penalizes forecast variability: an ensemble with systematically close-to-average spread will be less likely to produce events with large CRPS, so it will have a better (smaller) CRPSf. By design, it will also produce less extreme probabilities (this is quite clear in Fig. 2), so it will have a reduced capability for detecting extreme events. The performance metrics used in this paper tend to reward average skill. It would be useful to see metrics that are more sensitive to the detection of anomalous events, such as CSI or F1, F2, or the areas under the ROC and precision-recall curves for high physical thresholds.

This is a very good point. We agree that this is a drawback of the CRPSf. The CRPSf was defined here as a counterpart of the CRPSp to check that a better CRPSp does not hide some very bad forecasts. The CRPSp and CRPSf should be considered together. We will make it explicit in the text after the definition of the CRPSp and CRPSf.

We agree that our metrics favour average skill and that the study of anomalous events would be interesting. However, the low number of samples (90 forecasts) makes it difficult to investigate extremes. We investigated the prediction of the lowest 10th percentile of the 2m temperature and the highest 90th percentile of the 10m wind speed. Preliminary results show consistent behaviour with precedent results (see Figure below). We thus did not add it in the article. We will justify our choice at the beginning of Section 3.2: "*Due to the low number of samples in our case study (90 simulations), we do not investigate the prediction of extreme events*".

[Figure]

Figure: Brier Skill Score for the prediction of the 2m temperature 10th percentile.

[Figure]

Figure: Brier Skill Score for the prediction of the 10m wind speed 90th percentile.

Table 2: the formulae are unnecessarily heavy. For instance, the double sum over i and j, factor wi etc could be avoided by simply stating that the scores are spatially averaged. This is standard practice in the weather community.

We will modify Table 2 as you suggested. Thank you for your suggestion, the Table is now easier to read.

Fig 4, 5, 6: Some graphical indication of significance of the differences is needed, particularly the differences that are presented as results in the text. In maps, an effective method is to leave blank (white) the areas where SSR does not significantly differ from one.

Figure 4: For readability, the results of the statistical tests are shown in Appendix D. We will add better references to this appendix when discussing them in the text.

**Figure 5: We will update Figure 5 to marked grid points that are statistically significant.**

[Figure]

Figure 5. Maps of the differences between ECMWF and GaussW2 barycenter in terms of (top) CRPSS, (middle) CRPSp, and (bottom) CRPSf averaged over weeks 3 and 4. Green indicates that GaussW2 barycenter performs better and pink that ECMWF does. For readability reason, a discrete colorbar is used for the $\Delta$CRPSp and $\Delta$CRPSf which are more pixelized. Dots indicate grid points where the difference is statistically significant at the 5% level (using a two-sided Wilcoxon signed rank test for the CRPS and a McNemar test for the CRPSp and CRPSf).

[Figure]

Figure 6: The grid points for which the MSE is significantly different from the variance will be marked by a dot.

Figure 6. Spread-Skill-Ratio (SSR) for the weekly 2m temperature (left), 10m wind speed (center) and geopotential height at 500hPa (right) averaged over weeks 3 and 4 for the four single-model ensembles and the two barycenters. Orange indicates overdispersion (i.e. underconfident forecasts), and blue under-dispersion (i.e. overconfident forecasts). Dots indicate grid points where the ensemble variance and MSE of the ensemble mean are are statistically different at the 5% level (using a two-sided Wilcoxon signed rank test on the distribution of variance and MSE).

I.367: explain -> explained

This typo will be corrected.

I.373: nonexistent section number 22

The formatting will be corrected. The section number should be 2.2.2.

I.374: please provide references to studies where a similar ensemble is proven to be over-dispersed indeed.

We agree that ensembles suffer generally from under-dispersion in weather forecasting. The cases of over-dispersion are not common (but can happen, see Eade et al. (2024), Strommen et al, (2023) or Zhou et al. (2022)). This was meant as a general comment on the type of situation for which the GaussW2 would be an advantage, in symmetry to l370 for the L2 barycenter. However, we see how it can be misleading and we will reformulate that paragraph.

References:

Eade, R., D. Smith, A. Scaife, E. Wallace, N. Dunstone, L. Hermanson, and N. Robinson (2014), Do seasonal-to-decadal climate predictions underestimate the predictability of the real world?, *Geophys. Res. Lett.*, 41, 5620–5628, doi:10.1002/2014GL061146.

Strommen, K., MacRae, M., & Christensen, H. (2023). On the relationship between reliability diagrams and the "signal-to-noise paradox". *Geophysical Research Letters*, 50, e2023GL103710. https://doi.org/10.1029/2023GL103710

Zhou, X., and Coauthors, 2022: The Development of the NCEP Global Ensemble Forecast System Version 12. *Wea. Forecasting*, **37**, 1069–1084, https://doi.org/10.1175/WAF-D-21-0112.1.

I.376,383: nonexistent section number

The formatting will be corrected. The section numbers now appear as 4.1.

I.383: "better" than what ? L2 or SMEs ?

The comparison was with the L2 barycenter. This sentence will be reformulated into two sentences to improve readability and avoid such confusion. It will become "*Specifically, the L2 and GaussW2 barycenters have similar CRPSS and CRPSf, but the GaussW2 barycenter has a significantly better CRPSp (at the 5% level). Both barycenters also outperform the three SMEs for all metrics.*"

I.386-387: cumbersome sentence. Please rephrase.

The sentence will be rephrased for better readability: "*A particularly striking example is the CRPSS of the 10m wind speed: while the three SMEs show little to no skill, with very low or*

*even negative CRPSS, their barycenters have CRPSS similar to ECMWF's."*

I.402-405: I do not understand this statement: where do you see the influence of the forecast initial condition in eq.7 ?

We interpreted the variance of the input ensembles as the representation of their uncertainty due to initial conditions. We will rewrite this passage to make our hypothesis and interpretation clearer:

*"If we assume that the variances of the input ensembles represent the uncertainty due to their initial condition, their average in the formula of the L2 barycenter variance (first term of Eq. (7)) can be interpreted as a measure of the MME's initial-condition uncertainty. The second term, the variance of the input ensemble means, can be seen as a measure of the model uncertainty as it quantifies the differences between ensemble (means).*"

I.425: "use _of_"

This typo will be corrected.

I.455: add "model _systematic_" error. Some model errors are not systematic and thus cannot be claimed to be handled by the GaussW2 barycenter method. The claim can only be made for those model errors that are identically activated in all ensemble members of each SME, and that are persistent in time at any given point.

Thank you for pointing it out. Statistical methods like the barycenter-based MME can indeed only correct the model systematic error (and not the random one). "model error" will be replaced by "model systematic error".

I.461: same remark as on l.402: these methods do not separate initial condition errors from model errors in a general sense, such a separation would only occur under some simplifying assumptions that need to be more explicit in the text.

We will make our assumption clearer in l.402-405. We will also make our statement more general here by replacing "*initial conditions and model uncertainty*" by "*forecast uncertainty*".

Appendix B. this appendix does not seem relevant to the rest of the article. It should be deleted if it is not explicitly used in the text body.

Appendix B is referred to in line 302 when discussing the link between the L2 barycenter and the CRPS.

Tables D1 to D4: these are not very readable. A graphical presentation of the useful information would improve readability. "models are significantly different" from what ? From each other, or is L2-bary different from GaussW2-bary ?

We agree that these tables are hard to read. We tried to use a more graphical representation (see Figure below). While it would be more visually appealing in a presentation, we fear that in writing the unusual format would raise more questions than it would help the readability.

We have not found another satisfactory graphical representation. Thus, we will update the Table's captions to make more clear that we are testing if the models (in row versus in column) are statistically different from each other.

[Figure]

Figure: Attempt at graphical representation of Table D1.

---

## Author Comment (AC2)

The authors propose the Wasserstein barycenter for use in multi-model ensemble forecasting and test its relative performance against the more traditional L2 barycenter. Overall, the study is well-written and presented in a way that was digestible by a general audience, despite some of the more complicated mathematical constructs employed. While the results indicate that the Wasserstein barycenter does provide some advantages, is it not universally better than the traditional approaches. Nonetheless, this work is important for exploring how other options for ensemble forecasting could provide additional insights. I recommend the paper for publication subject to minor clarifications and revisions below.

We thank anonymous reviewer #2 for their overall assessment, the insightful comments and encouraging review. Indeed, in no way do we claim that the Wasserstein barycenter is universally advantageous, but rather argue that it is of interest as an alternative tool to explore and manipulate multi-model ensembles. Please find below our point-by-point responses:

Clarifications:

 1.  Line 164: "Note that they additionally assumed the signal to be stationary and periodic which we did not do." Can you clarify the implications of this choice?

The assumption that the signal is stationary and periodic allows them to simplify the computation of the Wasserstein mapping and barycenter. These hypotheses were too strong to be applied to our data set. The sentence will be modified as follows: "*Note that they additionally assumed the signal to be stationary and periodic to simplify the computations, however, these hypotheses are not adapted to our dataset*".

 2.  Line 201: Since the members of the Wasserstein barycenter isn't drawn from the set of possible simulated states, is it possible that it is not actually physically realizable? Namely, that there's no physically consistent way to connect the final and initial states?

This is an interesting question. The members of the L2 barycenter are taken from the initial ensembles and are thus physically consistent (according to the numerical weather model they came from as you noted in point 4 below). While the W2 barycenter allows for more flexibility in the sense that its members are not coinciding with the input ensembles, it indeed does not guarantee their physical consistency. However, the W2 barycenter does not treat the variables independently, their covariances are taken into account. This is explicitly done in the computation of the Gaussian Wasserstein barycenter (see equation l170).

In our case study, the barycenters are applied to several lead times of the same parameter (2m temperature, 10m wind speed or geopotential height).  Only looking at one parameter, it is difficult to check the physical consistency between several weekly values.

3.  Line 247: "Due to model errors, forecasts tend to drift away from the observed climate toward the model climatology as lead time increases." Isn't the drift also caused by initial observational uncertainty?

Forecast errors are indeed due to both model error and uncertainty on the initial condition. However, here, we were referring to the drift toward the model climatology, which is independent from the initial states and has a large impact at sub-seasonal scales.

Thank you for the question. We will clarify this in text as follows :"*Ensemble forecasts suffer from systematic errors (e.g. mean bias) due to uncertainties in the ensemble initialization and model formulation. In particular, forecasts tend to drift away from the observed climate toward the  model climatology as lead time increases (Takaya et al. 2019), making statistical correction essential at extended-range time scales*".

4.  Perhaps the fact that the W2 barycenter could give results which are not part of the ensemble is an advantage: ultimately every model cannot represent physical co-variances correctly and have their own biases (all modeled physical processes are approximations). By choosing a trajectory that is not present in the ensemble you are no longer bound to those biases.

The Wasserstein barycenter also depends on the variable covariances because they are derived from the model outputs in the GaussW2 barycenter. However it is true that the covariances are modified/merged by it and this allows having samples that could have not been generated by the individual models.

Thank you for raising these points (2 and 4), we will modify section 2.2.3 (including l 201) to reflect this discussion.

Minor comments:

1. Section numbers referred to in the manuscript seem to be rendered incorrectly by the latex template.

Thank you for pointing it out. The section numbers will be corrected in the revised version.

2. Line 143: Should the variables inside the curly brackets be a_1 and a_2 instead of a and b?

You are correct. Thank you for pointing it out, it will be corrected in the revised version.

3. Line 163: Parenthetical citation should be used for Flamary et al. (2020)

Parentheses will be added to this citation in the revised version.

4. Line 182: hypothesis hypothesis

This typo will be corrected in the revised version.

5. Line 428: multi-model

This will be corrected in the revised version.